# Novel insights from a multiomics dissection of the Hayflick limit

Michelle Chan, Han Yuan†, Ilya Soifer†, Tobias M Maile, Rebecca Y Wang, Andrea Ireland, Jonathon J O'Brien, Jérôme Goudeau, Leanne JG Chan, Twaritha Vijay, Adam Freund, Cynthia Kenyon, Bryson D Bennett, Fiona E McAllister, David R Kelley, Margaret Roy, Robert L Cohen, Arthur D Levinson, David Botstein*, David G Hendrickson*

Calico Life Sciences, LLC, South San Francisco, United States

**Abstract** The process wherein dividing cells exhaust proliferative capacity and enter into replicative senescence has become a prominent model for cellular aging in vitro. Despite decades of study, this cellular state is not fully understood in culture and even much less so during aging. Here, we revisit Leonard Hayflick's original observation of replicative senescence in WI-38 human lung fibroblasts equipped with a battery of modern techniques including RNA-seq, single-cell RNA-seq, proteomics, metabolomics, and ATAC-seq. We find evidence that the transition to a senescent state manifests early, increases gradually, and corresponds to a concomitant global increase in DNA accessibility in nucleolar and lamin associated domains. Furthermore, we demonstrate that senescent WI-38 cells acquire a striking resemblance to myofibroblasts in a process similar to the epithelial to mesenchymal transition (EMT) that is regulated by t YAP1/TEAD1 and TGF-β2. Lastly, we show that verteporfin inhibition of YAP1/TEAD1 activity in aged WI-38 cells robustly attenuates this gene expression program.

## Editor's evaluation

This manuscript reports a unique, comprehensive, multi-omic resource for the study of replicative senescence. This resource encompasses temporal metabolomic, proteomic, bulk transcriptomic, single cell transcriptomic, and chromatin accessibility states of fibroblasts as they transition from proliferative to replicatively senescent. Hence, it will be considered a valuable resource by aging researchers.

*For correspondence:
botstein@calicolabs.com (DB);
dgh@calicolabs.com (DGH)

†These authors contributed equally to this work

## Introduction

Replicative senescence in animal cells growing in vitro was first discovered by Leonard Hayflick. He found that primary human diploid fibroblast cell lines ceased to proliferate after an extended number of serial passages (*Hayflick, 1965*). Since then, considerable work has been done to describe this phenomenon. A major causal feature of replicative senescence is telomere erosion, a process in which the telomeres gradually shorten with increasing cellular divisions. Eventually, the telomeres become uncapped which triggers a DNA damage response that results in cell cycle exit (*Harley et al., 1990*). It is understood that this is due to the absence of the telomerase reverse transcriptase (hTERT), the catalytic component of human telomerase which adds telomeric sequences to the ends of chromosomes to maintain telomere length in germ cells and stem cells. As hTERT activity is undetectable in normal human somatic cells, telomere attrition is a common aging phenotype hypothesized to underlie cellular senescence at the organismal level (*López-Otín et al., 2013*; *Meyerson et al., 1997*). Replicative senescence of human somatic cell lines in vitro can be avoided by overexpression of hTERT

**Figure 1.** Expression dynamics of replicative senescence (RS), radiation induced senescence (RIS), increasing cell density (CD) and hTERT WI-38 cells. (**A**) Experimental design for the RS, RIS, CD and hTERT experiments. (**B**) Days in culture vs. population doublings (PDL) for WT WI-38 cells (red) and hTERT immortalized cells (green). Labeled points denote sample collection and are expressed in PDLs for RS time course or PDL controls (PDL.ctrl) for the hTERT time course. Temporally paired samples indicated by vertical dotted lines. PDLs 52 and 53 were collected in a separate 'deep' senescence time course. (**C**) Percent cells staining positive for SA-β gal staining (y-axis) at increasing PDLs (x-axis) for WI-38 cells (red) and hTERT (green). Points represent replicate values. (**D**) Scatterplot of log2 fold changes in gene expression for PDL 50 vs. PDL 20 (x-axis) vs. senescence log2 fold change derived from a generalized linear model compiling gene expression changes across multiple fibroblast cell lines during replicative senescence (y-axis) (*Hernandez-Segura et al., 2017*). (**E**) Hierarchical clustering of significant gene expression changes (FDR adjusted p-value < 0.01) across all conditions (n=3 replicates). Values are log2 fold change vs. the average of the first time point of each condition. Reference time point not shown. (**F**) Significant (FDR adjusted p-value < 0.01) Gene Set Enrichment Analysis (GSEA) results for RS, RIS, and CD using the MSigDB Hallmarks annotation set. The -log10 p-value is colored by direction of enrichment (red=up, blue=down).

*Figure 1 continued on next page*

*Figure 1 continued*

The online version of this article includes the following source data and figure supplement(s) for figure 1:

**Source data 1.** RNA-seq quality control metrics.

**Source data 2.** RNA-seq Gene expression data and differential analysis.

**Source data 3.** RNA-seq GSEA.

**Source data 4.** RNA-seq GSEA for indivdual timepoints.

**Figure supplement 1.** Sample manifest.

**Figure supplement 2.** Senescence markers.

**Figure supplement 3.** Senescence markers cont'd.

**Figure supplement 4.** Individual time point GSEA.

**Figure supplement 5.** Pilot experiment for WI-38 husbandry.

which prevents telomere shortening and confers apparently unlimited replicative capacity (*Bodnar et al., 1998*).

Beyond growth arrest and telomere shortening, phenotypic changes exhibited in replicatively senescent cells include the senescence associated secretory phenotype (SASP). Proteins of the SASP include proinflammatory cytokines, growth factors, angiogenic factors, and proteases. The SASP has been shown to play a role in paracrine signaling whereby senescent cells can promote local wound healing and/or drive healthy neighboring cells into senescence (*Acosta et al., 2013*; *Coppé et al., 2006*; *Demaria et al., 2014*; *Coppé et al., 2008*). Replicatively senescent cells also accumulate DNA and protein damage, accumulate lipids, and lose regulatory control of mitochondria and lysosomes (*Gorgoulis et al., 2019*).

The phenotypic similarity (telomere attrition, epigenetic alterations, mitochondrial dysfunction, and loss of proteostasis) between the cell autonomous aging hallmarks and in vitro senescence has led to the hypothesis that senescent cells in vivo play a causal role in organismal aging and aging-related diseases (*Hernandez-Segura et al., 2018*; *Campisi and d'Adda di Fagagna, 2007*; *Sedelnikova et al., 2004*; *Jeyapalan and Sedivy, 2008*; *Herbig et al., 2006*; *Childs et al., 2015*; *López-Otín et al., 2013*).

Consistent with this model, several age-related disease states can be directly linked to telomere length or telomerase activity. For example, up to 15% of familial idiopathic pulmonary fibrosis (IPF) cases arise from mutations in telomerase and up to 25% of sporadic cases occur in people with telomere lengths less than the 10th percentile. (*Tsakiri et al., 2007*; *Armanios et al., 2007*; *Cronkhite et al., 2008*; *Alder et al., 2008*; *Stuart et al., 2015*; *Raghu et al., 2006*; *Duckworth et al., 2021*; *Stuart et al., 2014*; *Dai et al., 2015*). Furthermore, the elimination of senescent cells in a number of age-related diseases, such as cardiac fibrosis, pulmonary fibrosis, neurodegenerative diseases, osteoporosis, and metabolic disorders have been argued to alleviate the disease state (*Pignolo et al., 2020*). Clinical trials for senolytics targeting fibrotic diseases, osteoporosis, frailty, and metabolic syndromes are currently underway (*Borghesan et al., 2020*).

The mechanism by which senescent cells might contribute to aging phenotypes is currently still unclear (*Hernandez-Segura et al., 2017*; *Hernandez-Segura et al., 2018*). In an effort to bring clarity to the replicative senescence process in vitro that in turn could elucidate in vivo function, we revisited and redesigned the original Hayflick experiment. Making use of recent advances in technology we tracked changes in cell state throughout the rplicative lifespan of the original Hayflick WI-38 cell line using bulk RNA-seq, single-cell RNA-seq (scRNA-seq), ATAC-seq, metabolomics and proteomics.

Overall, our data recapitulate many known features of the in vitro senescence process while simultaneously providing novel insight. First, the time resolution of our experiment coupled with single-cell trajectory analysis reveals that senescence is a gradual process that shares transcriptional, proteomic, and metabolomic features with epithelial-mesenchymal transition (EMT). Second, our metabolomic data identifies Nicotinamide N-methyltransferase (NNMT) activity as a potential initiating event in replicative senescence dependent loss of epigenetic silencing. Third, we show that these genomic regions that exhibit increased accessibility with increasing cellular age are concentrated in nucleolar/lamin associated domains and correspond with observed changes in the replicative senescence transcriptome. Lastly, integration across data modalities reveals that senescent WI-38 cells bear a strong

resemblance to myofibroblasts. We provide bioinformatic and experimental evidence that the YAP1/TEAD1 transcription factor complex and TGF-β2 signaling are putative regulators of the transition to this state. Together our data suggests that a process similar to fibroblast to myofibroblast transition (FMT; analogous to EMT) is an intrinsic aspect of the replicative senescence phenotype in WI-38 fibroblasts.

## Results

## Transcriptomic profiles of replicative senescence, radiation-induced senescence, and increasing cellular density

To capture the replicative senescence process with high resolution we designed an experiment to continuously grow and intermittently sample cells from a starter batch of WI-38 cells that had undergone only 20 population doublings (PDL 20) (methods). To distinguish between replicative senescence-dependent changes and those arising from altered cell density and growth rate, we performed a cell density control study (methods). Briefly, early PDL cells were sampled at increasing levels of cell density to measure gene expression changes associated with cellular density and decreasing cell proliferation independent of replicative senescence (*Figure 1A*). In addition to the cell density control, we also included TERT immortalized WI-38 cells (hTERT) grown in parallel and sampled alongside wild type (WT) cells as a control for long term culturing and day-to-day sampling batch effects (*Bodnar et al., 1998*). The hTERT cell line was generated in advance of the experiment. Frozen aliquots of the hTERT line were thawed alongside and sampled in parallel with WT WI-38 cells. As expected, hTERT immortalized cells grew at a constant rate and did not slow or cease growth (*Figure 1B*).

We also included a radiation induced senescence condition to test for differences between replication and radiation induced senescence and to isolate changes arising from acute DNA damage response. Finally, we sampled proliferating WT and hTERT WI-38 cells at multiple PDLs for RNA-seq, scRNA-seq, proteomics, metabolomics and ATAC-seq until the WT WI-38 cells had reached senescence as measured by the cessation of growth (*Figure 1B*, *Figure 1—figure supplement 1B*).

We first examined the bulk RNA-seq data to compare and contrast replicative senescence with radiation induced senescence, cell density, and hTERT cells (quality control metrics located in *Figure 1—source data 1*). Differential gene expression analysis using RNA-seq revealed 6955 genes change with increasing PDL, 7065 genes change in response to ionizing radiation, and 9958 genes vary with increasing cell density (FDR adjusted p-value < 0.01, *Figure 1—source data 2*). Notably, the transcriptional changes we observed were consistent with previous studies. *Figure 1D* shows the strong correlation of the pattern of induced transcription established in our PDL 50 cultures and the fibroblast derived senescence-associated signature compiled across multiple fibroblast senescence gene expression experiments from cell lines derived from different tissue depots (*Hernandez-Segura et al., 2017*). We also observed induction of senescence-associated-β galactosidase activity (*Figure 1C*) as well as p16 and p21 in both our RNA-seq, proteomics data, and via western blot (*Figure 1—figure supplement 2*, *Figure 1—figure supplement 3*). These data reveal that the cells in our time course display classic features of senescent cells and exhibit transcriptomic changes that are highly correlated with previous studies of replicative senescence. Together, these data strongly suggest that our WI-38 cells successfully reached replicative senescence.

Hierarchical clustering of all significantly changing genes across the four conditions highlighted several important features of our experiment. First, as expected, gene expression in the hTERT-immortalized WI-38 cells remained largely stable. Second, replicative senescence, radiation induced senescence, and cell density exhibited many shared, but also unique, gene expression changes with respect to both the identity of differentially expressed genes and the magnitude of their changes (*Figure 1E*).

To facilitate a biological interpretation of these shifting transcriptomic landscapes, we applied gene set enrichment analysis (GSEA) using the MSigDB Hallmark annotation sets to learn which general processes are shared and distinct between replicative senescence, radiation induced senescence, and cell density (*Figure 1F and Figure 1—source data 3*; *Liberzon et al., 2015*; *Subramanian et al., 2005*; *Mootha et al., 2003*). All three perturbations (and not the immortalized cells), exhibited dramatic reductions in expression of genes belonging to S, G2M, and M cell cycle phases, consistent with cessation of cell division (*Figure 1F* - top cluster). Perhaps driven by this shift, we

also found significant overlap in additional enriched annotations among genes induced in replicative senescence, radiation induced senescence and cell density, albeit with some variation in significance. Many of the shared enriched annotations can be categorized as stress responses, for example apoptosis, p53 pathway, inflammatory and interferon responses, and STAT3/5 signaling. In addition, we also observed enrichment of several developmental gene sets including myogenesis, angiogenesis, and adipogenesis.

We observed relatively few gene sets with discordant patterns across the replicative senescence, radiation induced senescence, and cell density experiments (*Figure 1E*). However, the gene set for epithelial to mesenchymal transition (EMT) was significantly enriched in genes that increased with replicative senescence as opposed to radiation induced senescence (Benjamini-Hochberg corrected p=5.5e-6 vs. Benjamini-Hochberg corrected p=0.12) or cell density wherein the term was actually underrepresented, but not significantly so (Benjamini-Hochberg corrected p=0.4). Although replicative senescence, radiation induced senescence, and cell density are highly similar at the abstracted level of enriched gene sets, it is clear in *Figure 1E* that there are many induced genes specific to replicative senescence. In addition, we applied GSEA to each individual time point for each condition (*Figure 1—figure supplement 4*, *Figure 1—source data 4*) and found the EMT gene set is enriched early and robustly during replicative senescence. Thus, the EMT gene set appears to represent a particularly important aspect of replicative senescence biology.

## Single-cell RNA-Seq reveals that replicative senescence is a gradual process

Interestingly, the vast majority of gene expression changes evident by PDL 50 begin to manifest at much earlier PDLs. This observation is consistent with two distinct possibilities: (A) The senescence expression changes accrue early and gradually in the majority of cells without respect to proliferation status, or (B) the bulk RNA-seq profiles are a changing admixture of transcriptionally distinct mitotic, G1, and senescent cells. Both models have substantial support in the literature (*Tang et al., 2019*; *Wiley et al., 2017*; *Nassrally et al., 2019*; *Smith and Whitney, 1980*; *Passos et al., 2007*; *Smith and Whitney, 1980*). To discriminate between these possibilities, we employed single-cell RNA-seq (scRNA-seq) to directly measure percentages of senescent and cycling cells.

Briefly, we collected between 1000 and 2000 cells at increasing PDLs of WI-38s and matched hTERT time points at beginning and end of time course. We applied a modified (see methods) 10 X Genomics protocol to capture and preserve the more fragile senescent cells (*Figure 2—figure supplement 1*). We profiled a total of 11,000 cells (9000 WT and 2000 hTERT). When aggregated into pseudo-bulk expression profiles (i.e. the summation of all cells), the single cell results are highly correlated with expression changes observed in bulk RNA-seq (r=0.81), (*Figure 2—figure supplement 2*).

We first classified cells as either S, G2M, or G1 phase using canonical markers for the S and G2M phases as previously described *Nestorowa et al., 2016*. As expected, based on bulk transcriptomics, the number of cells in either S or G2M decreases with increasing PDL, whereas the cycling cell proportions are stable in the hTERT timepoints (*Figure 2A and B*).

Next, we projected all WT cells and two hTERT time points together to identify broad patterns in replicative aging PDL cells versus cycling hTERT cells. To do this, we employed UMAP projection as opposed to t-SNE as the former is faster and preserves local structure more so than the latter (*McInnes et al., 2018*). Cells from the two cell lines organize separately, however, share a similar geometry composed of a S/G2M phase roundabout and a large G1 lobe (*Figure 2C*). Although the overall pattern of WI-38 cells for both cultures was highly similar, we observed progression of cell grouping with PDL that had no concordance with increasing hTERT time points (*Figure 2D* - red to blue). Specifically, PDL 50 cells (but not the temporally paired hTERT PDL.ctrl 50 time point) organize apart from all other cells within the UMAP projection. To test if these cells were senescent, we scored all cells with a senescent gene expression signature derived from the bulk RNA-seq and found that the highest scoring cells reside within this grouping composed of PDL 50 cells (methods) (*Figure 2—figure supplement 4*).

To test model A vs. B, we compared the pseudo-bulk gene expression profiles across the cell cycle at each PDL. We reasoned that if replicative senescence were gradual, we would observe the replicative senescence gene expression pattern even in young cycling cells without respect to G1/S/G2M. Indeed, the replicative senescence signature is present in all cell cycle phases even in early PDLs

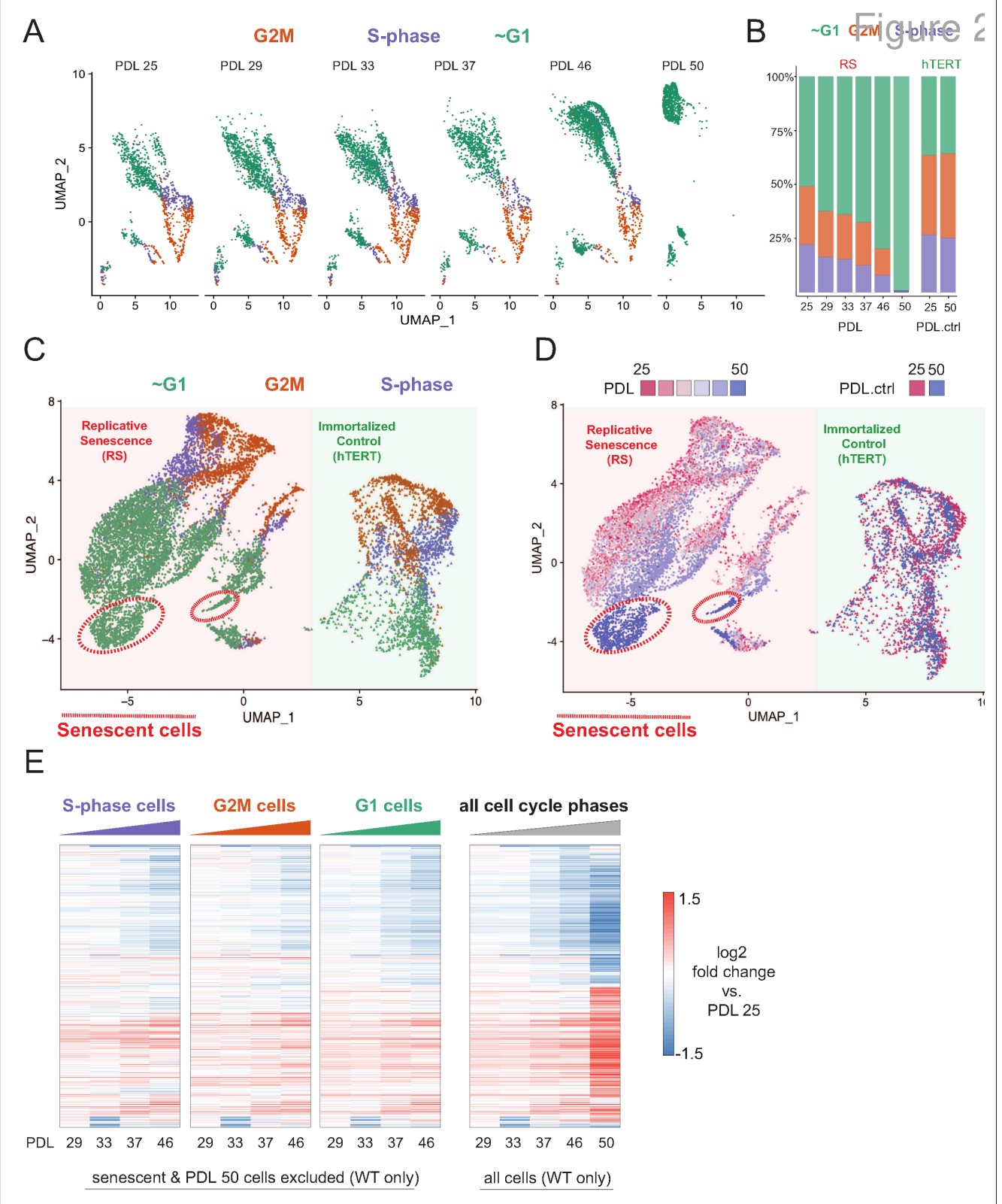

**Figure 2.** Cell cycle exit and distribution on approach to replicative senescence (RS) does not explain gradual increase in the replicative senescence transcriptome. (**A**) Individual UMAP projections of WT WI-38 cells; PDL colored by phases of the cell cycle (G1 = green, G2/M = orange, S-phase = purple). (**B**) Bar graph of cell cycle state percentages defined by transcriptomic score (y-axis) by PDL (x-axis) for WT WI-38 cells (left) and hTERT WI-38 cells (right). (**C**) UMAP projections of WT and hTERT WI-38 cells by PDL and PDL.ctrl colored by different phases of the cell cycle defined by

*Figure 2 continued on next page*

*Figure 2 continued*

transcriptomic score. (**D**) UMAP projection of all WI-38 cells. Increasing PDLs (RS) or PDL control (hTERT) are colored from early (red) to late (blue). (**E**) The replicative senescence transcriptome manifests early in all phases of cell cycle. Heatmap of hierarchical clustering of gene expression values of differentially expressed genes as aggregated transcriptomic profiles for each cell cycle phase and PDL (left) versus all cells (right). Values are log2 fold change of each PDL against the reference PDL 25 (not shown).

The online version of this article includes the following figure supplement(s) for figure 2:

**Figure supplement 1.** Buffer optimization for scRNA-seq.

**Figure supplement 2.** Correlation between bulk and single cell RNA-seq.

**Figure supplement 3.** UMAP of mitotic cells.

**Figure supplement 4.** Senescence and cell density scoring of scRNA-seq cells.

**Figure supplement 5.** mages of cell density at sampling.

(*Figure 2E*- left panels vs. far right panel). In addition, bioinformatic segregation and UMAP projection of cells scored as S/G2M phase cells revealed that the dominant source of variance in cycling cells is the PDL (*Figure 2—figure supplement 3*). Lastly, we did not observe any cells < PDL 50 in close association with the senescent cells in the UMAP projection PDLs (*Figure 2C and D*).

It has previously been reported and observed here (*Figure 2—figure supplement 5*) that cell size increases with PDL in primary cell lines (*Ogrodnik et al., 2019*). We hypothesized that perhaps the higher PDL cells by nature of their larger size, might reach contact inhibition faster than early PDL cells. In this scenario, given the high correlation between the gene expression signatures of senescence and cell density, the increase in senescence signature could be an artifact of cell culture arising from changing morphology. To test this possibility, we scored our single cells again with modified signatures for senescence and cell density composed only of genes uniquely induced in either perturbation (*Figure 2—figure supplement 4*). We found that the specific senescent signature increases with PDL in WT WI-38 cells in contrast with the specific cell density signature which stays relatively flat. These data provide more evidence supporting a gradual increase in the senescent gene expression program that is independent from changes in cell density.

Together, these results argue in favor of a gradual model (A) of replicative senescence wherein cells ramp up expression of the replicative senescence program with increasing PDL even when still proliferative. We cannot rule out there are a small minority of cells that enter senescence early in the experiment as previously reported (*Smith and Whitney, 1980*; *Passos et al., 2007*). However, for the cells that made it to PDL 50, we observed that the senescent gene expression program manifests in their primogenitors. These data are also consistent with the previously reported changes in cellular phenotype (larger cell size and increased cycling time) with increasing PDL (*Macieira-Coelho and Azzarone, 1982*; *Ogrodnik et al., 2019*; *Neurohr et al., 2019*; *Absher et al., 1974*). Importantly, this observation suggests that aspects of cellular senescence are present in non-senescent cells. This result raises the intriguing possibility that the reported pathological features of senescent or senescent-like cells in vivo might also manifest in cells that are not classically senescent.

## Proteomic landscape of replicative senescence WI-38 cells suggests large metabolic alterations

We next turned to our proteomics data generated from the same cultures and time points as above. We obtained high confidence measurements for ~8500 proteins (*Figure 3—figure supplement 1*, *Figure 3—source data 1*). Similar to our transcriptional results, the hTERT samples did not exhibit large changes in the proteome with experimental progression (*Figure 3—figure supplement 1*). Overall, we observed high concordance between transcript and protein levels with very few outliers (r=0.71 for PDL 50 vs. PDL 20, *Figure 3—figure supplement 2*). We employed GSEA using hallmark annotations and observed a sharp depletion of proliferation and mitosis associated gene sets supporting our previous findings (*Figure 3A and Figure 3—source data 1*). Likewise, the proteomics data recapitulated enrichment of the EMT annotation set along with the adipogenesis and myogenesis.

Multiple enriched sets pointed to replicative senescence-dependent shifts in cellular energy utilization (*Figure 3A*). Of note, we found that although under-enriched in our replicative senescence transcriptome (*Figure 1F*), The oxidative phosphorylation hallmark was highly enriched in the proteomics data. We plotted the oxidative phosphorylation genes driving the set enrichment in *Figure 3—figure*

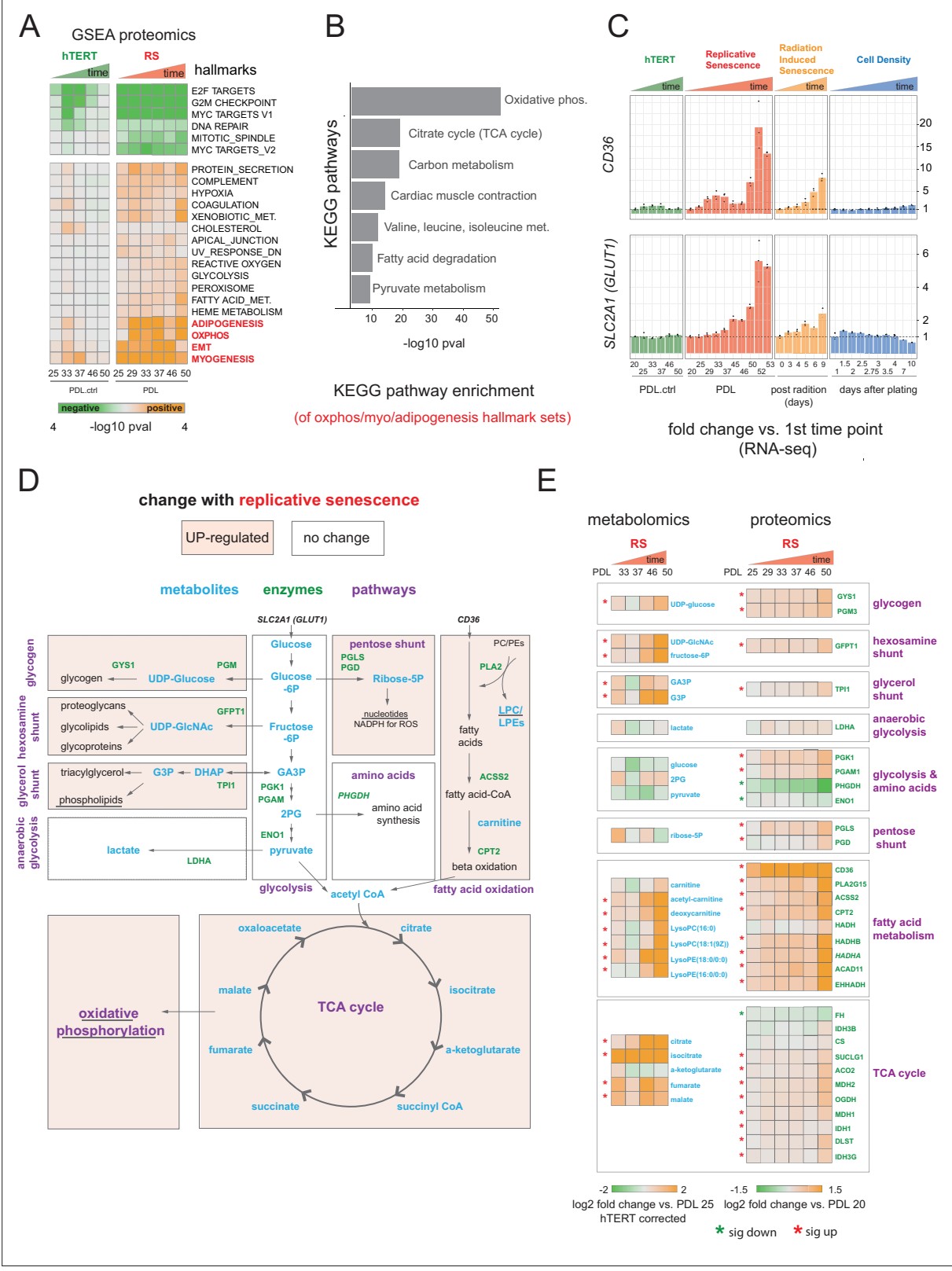

**Figure 3.** Proteomic and metabolomic changes during replicative senescence. (**A**) Significant (Benjamini-Hochberg adjusted pvalue < 0.01) GSEA results for protein changes (n=3) using the MSigDB Hallmarks. Reference time point not shown. Values are -log10 p-value and are colored by the direction (orange=up, green=down). (**B**) KEGG and GO term enrichment of genes from A for selected annotations. (**C**) Gene expression changes (n=3) for insulin-independent glucose transporter *SLC2A1* (left) and fatty acid transporter *CD36* (right) in the hTERT, replicative senescence, radiation

*Figure 3 continued on next page*

*Figure 3 continued*

induced senescence and cell density time courses. (**D**) Diagram of glycolysis, glycolytic shunts, fatty acid import and oxidation, TCA cycle/oxidative phosphorylation. Metabolites are blue, proteins are green. (**E**) Metabolite (n=4) and protein changes (n=3) with senescence from D. Average corrected metabolite log2 fold change vs. PDL 25 as reference (not shown) are plotted. Proteomics values are average log2 fold change vs. PDL 20 as reference (not shown). *:Significant changes (FDR adjusted p-value < 0.05 metabolomics, FDR adjusted p < 0.01 proteomics).

The online version of this article includes the following source data and figure supplement(s) for figure 3:

**Source data 1.** Proteomics quantification, differential analysis and GSEA.

**Source data 2.** Metabolomics quantification and differential analysis.

**Figure supplement 1.** PDL-dependent changes in the senescent proteome vs. hTERT cells.

**Figure supplement 2.** Correlation between bulk RNA-seq and proteomics.

**Figure supplement 3.** Proteomic changes in oxidative phosphorylation annotation.

**Figure supplement 4.** PDL-dependent changes in the senescent metabolome vs. hTERT cells.

**Figure supplement 5.** Kennedy pathway utilization during replicative senescence Kennedy Pathway diagram.

*supplement 2.* This projection revealed that many genes in this set fall into a quadrant (positive proteomic, negative transcriptomic) suggesting discordant regulation of these genes.

The oxidative phosphorylation hallmark set is composed of multiple closely linked mitochondrial complexes and functions that regulate cellular energy flux, for example TCA cycle, fatty acid oxidation, pyruvate metabolism, ATP synthase etc. We divided the oxidative phosphorylation gene set into into these constituent parts and visualized the changes in proteins with increasing PDL. We observed PDL-dependent increased in proteins from all categories except for mitochondrial assembly and structures. These results are consistent with altered mitochondrial function (*Figure 3—figure supplement 3*).

Next, we examined the annotations for the highest enriched hallmark sets from *Figure 3A* and found strong enrichment of multiple KEGG metabolic pathways. (*Figure 3B*). In addition, both the glucose transporter *SLC2A1* and the fatty acid scavenger *CD36* transcripts exhibited strong, early, and replicative senescence specific up-regulation suggesting altered cellular energy requirements (*Figure 3C*). Together these results point to a drastically altered replicative senescence metabolic landscape.

## Senescent WI-38 cells exhibit increased utilization of fatty acid metabolism and glycolytic shunts

To generate a metabolic profile for replicative senescence in WI-38 cells, we harvested samples at increasing PDLs alongside input material for all other data types (methods). From our metabolomics data, we identified 285 compounds (*Figure 3—figure supplement 4*, *Figure 3—source data 1*). To remove batch effects that were observed in several PDL and the matched hTERT samples, we corrected each PDL with its paired hTERT sample. We then calculated significant changes over time for each metabolite (FDR adjusted p < 0.05, *Figure 3—source data 2*). To further guide our metabolic analysis, we focused our queries based on the enriched metabolic pathways found in the proteomic analysis. Specifically, we studied the changes to glycolysis, oxidative phosphorylation and fatty acid metabolism (*Figure 3D*).

Following the glucose and the increased expression of the glucose importer *SLC2A1*, we looked at the central stem of glycolysis as a potential source of energy fueling the altered senescent metabolism suggested by our proteomics data. (*Figure 3D and E*). We found that metabolites and enzymes dedicated to pyruvate generation and/or lactate production did not change in a concordant manner. We instead found that most of the glycolytic shunts exhibited upregulation at both the metabolite and protein level (*Figure 3E*). These results suggest an increase in allocation of glucose for manufacture of various biomolecules and their precursors (glycogen, hexosamines, phospholipids) alongside or in lieu of pyruvate or lactate production (*Figure 3E*).

Metabolites and enzymes involved in fatty acid import (e.g. *CD36*) and oxidation for the purpose of energy generation appear strongly up-regulated (*Figure 3E*). We also saw that metabolites involved in the de novo production of phospholipids via the Kennedy pathway are highly upregulated, specifically phosphotidylethanolamines precursors (*Figure 3—figure supplement 5*).

From these data, it is clear that WI-38 cells approaching replicative senescence undergo drastic shifts in metabolism. Specifically, we observed increased glucose utilization in glycolytic shunts coupled with an increase in fatty acid import and oxidation. It is possible that the senescent cells are switching to fatty acid oxidation to fuel increased TCA cycling and oxidative phosphorylation as glucose is diverted to macromolecule production. Indeed, metabolomic data collected from various types of senescence models has shown increased fatty acid oxidation, lipid accumulation, TCA up-regulation, and glycolytic alterations (*Zwerschke et al., 2003*; *Ogrodnik et al., 2017*; *Flor et al., 2017*; *Unterluggauer et al., 2008*; *Johmura et al., 2021*). Likewise, consensus metabolomic findings in EMT models report increased TCA cycle products, altered lipid metabolism, and activated hexosamine pathway (*Hua et al., 2020*; *Corbet et al., 2020*; *Lucena et al., 2016*). These data provide functional metabolic evidence supporting a connection between replicative senescence, metabolic rewiring, and the increased expression of EMT-related genes.

## Nicotinamide N-methyltransferase (NNMT) links nicotinamide adenine dinucleotide (NAD) and methionine metabolism as a putative heterochromatin regulator

The interplay between metabolism and the epigenome is an increasingly recognized mechanism through which nutrient availability shapes chromatin and regulates cell state (*Lu and Thompson, 2012*). The metabolites nicotinamide adenine dinucleotide (NAD+) and methionine metabolism power the deacetylation and methylation of histone tails and DNA required for maintaining repressive DNA conformations (*Lu and Thompson, 2012*). In a compelling intersection, perturbation of NAD, methionine, and heterochromatin levels have all previously been reported in multiple aging contexts including replicative senescence (*James et al., 2016*; *López-Otín et al., 2016*; *Kozieł et al., 2014*; *Yousefzadeh et al., 2020*; *Benayoun et al., 2015*). In review of our replicative senescence data for the metabolic and proteomic components of the NAD and methionine pathways, we found that one of the largest and earliest changes for any protein (or transcript) was the increased expression of Nicotinamide N-methyltransferase (NNMT) (*Figure 4A and B*).

DNA and histone methylation require abundant levels of the universal methyl donor S-adenosyl methionine (SAM). NNMT not only depletes SAM by catalyzing the removal of its methyl group, it does so by fusing the methyl group to the NAD+ precursor nicotinamide (NAM) resulting in the generation of methyl nicotinamide (MNA) (*Ulanovskaya et al., 2013*). Thus, NNMT effectively acts as a sink for two of the primary metabolites a cell requires for silencing chromatin and regulating gene expression (*Komatsu et al., 2018*). We found that MNA levels mirror those of NNMT; increasing early and robustly during replicative senescence progression consistent with high NNMT activity. Both SAM and is methyl-depleted counterpart, S-adenosyl homocysteine (SAH), were depleted with replicative senescence albeit to a lesser extent than the observed increase in MNA. NAMPT and NMNAT1, enzymes in the NAD salvage pathway, were weakly down-regulated at the protein level (*Figure 4A and B*).

High NNMT expression has been implicated in SAM depletion in embryonic stem cells and cancer-associated fibroblasts (CAFs) (*Eckert et al., 2019*; *Sperber et al., 2015*). In both cases, the functional consequences were similar; DNA hypomethylation and decreased capacity to form or maintain silenced loci and heterochromatin. The important point is that the loss of silencing is actively promoted through NNMT activity. One intriguing hypothesis is that increased NNMT activity with replicative senescence promotes loss of silencing and heterochromatin via SAM depletion (*Figure 4C*).

## Increased DNA accessibility and transcription from nucleolar and lamin-associated domains is a dominant feature of the replicative senescence epigenome

To study genome-wide changes in the epigenetic landscape of senescent cells, we collected ATAC-seq data during the replicative senescence time course. ATAC-seq libraries from all PDLs and hTERT PDL controls exhibited similar ATAC-seq fragment size distribution emblematic of nucleosomal patterning, high alignment rates, very low mitochondrial read percentages, and strong enrichment around transcriptional start sites. To enrich for nucleosome free regions of DNA, we applied size selection to our ATAC-seq libraries prior to sequencing (methods, *Figure 5—figure supplement 1*, *Figure 5—figure*

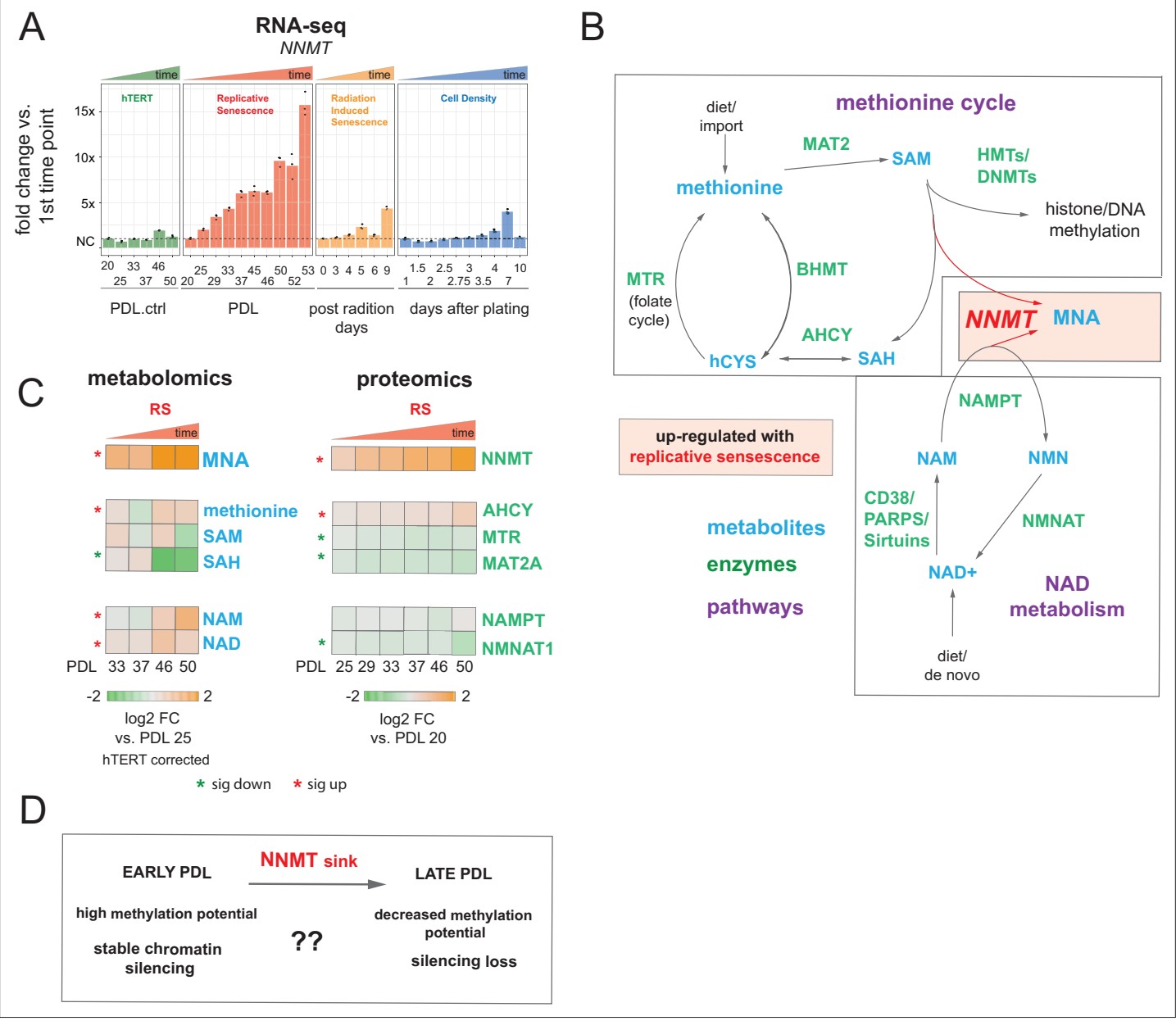

**Figure 4.** Nicotinamide n-methyltransferase (NNMT) links nicotinamide adenine dinucleotide (NAD) and methionine metabolism during replicative senescence. (**A**) Gene expression fold changes for the NNMT (top) in the hTERT, replicative senescence (RS), radiation induced senescence (RIS), and cell density (CD) time courses (three replicate average, each point a replicate). Heatmaps of metabolite and protein changes during replicative senescence from B (bottom-replicate average). (**B**) Metabolic diagram of the methionine and NAD salvage pathways. Metabolites are blue, proteins are green. Shading indicates inferred pathway direction during replicative senescence based on metabolite/protein changes in A. (**C**) Heatmaps of metabolite (four replicate average) and protein (three replicate average) changes with replicative senescence from B. Metabolomics data was batch corrected against hTERT samples and expressed as log2 fold change vs. PDL 25 as reference (not shown). Proteomics data expressed as log2 fold change vs. PDL 20 as reference (not shown). Significant changes (FDR adjusted p < 0.05 metabolomics, FDR adjusted p-value < 0.01 proteomics) during replicative senescence are denoted with asterisks. (**D**) Potential model for NNMT and metabolic regulation of chromatin state during replicative senescence.

*supplement 2*, *Figure 5—figure supplement 3* and *Figure 1—source data 1*). These quality control metrics are indicative of high-quality ATAC-seq libraries at all sampling time points and PDLs.

We next quantified the read distribution across chromatin states previously annotated for another human fetal lung fibroblast cell line (IMR-90, *Figure 5—source data 1*; *Ernst and Kellis, 2015*). We categorized 25 distinct states into four broad categories: promoters, enhancers, transcription, and

miscellaneous, the last of which is composed largely of heterochromatic and undefined states, for example H3K9me3 and H3K27me3-rich regions, zinc-finger repeats, and quiescent (defined by the absence of histone marks, accessibility, or gene features) which we refer to as 'undefined' (*Ernst and Kellis, 2015*).

In comparing the read distribution across chromatin states from hTERT immortalized cells and WT WI-38 cells, we observed an increase in ATAC-seq reads falling into the undefined chromatin regions at the relative cost to all other states (*Figure 5A-left panel*). We next plotted the read distribution across chromatin states as a function of increasing time for both cell lines and found these shifts to be associated with increasing PDL in the WT WI-38 cells; hTERT ATAC-seq read proportions across chromatin states remained stable over time (*Figure 5A-right panel*).

To determine if this shift in accessibility to undefined and heterochromatic states was indicative of increasing noise versus coherent changes in accessibility, we used the ATAC-seq reads to call peaks of accessibility. From the WT and hTERT ATAC-seq data from all time points, we identified 363,470 ATAC-seq peaks (*Figure 5—source data 2* and *Figure 5—source data 3*). We found a slight but significant (~5%, p=0.014,linear regression) linear PDL-dependent decrease in the fraction of reads in peaks during replicative senescence (*Figure 5—figure supplement 4A*, *Figure 5—source data 4*). We tested experimentally whether the increased heterochromatic accessibility was an artifact of increasing numbers of dead cells in later PDL samples by repeating the experiment using late (PDL 45) cells with a cross-linking agent (propidium monoazide-PMA) that renders DNA from dead cells inert as previously described (*Hendrickson et al., 2018*). No significant change in read distribution across chromatin states was observed with addition of PMA (*Figure 5—figure supplement 4B*).

We then divided the ATAC-seq peaks into each of the 25 discrete chromatin states and calculated the fold change in accessibility for each state and PDL compared to the first time point. We observed that changes in peak accessibility mirrors that for all reads. Again we observed a clear increase in accessibility with PDL across the miscellaneous category with a concomitant decrease in all other states in WT WI-38 cells and not in immortalized hTERTs (*Figure 5B*).

The undefined state from the miscellaneous category piqued our interest for two reasons, one being that it alone accounted for 20–40% of all ATAC-seq reads and a large 10% increase with senescence. The second point of interest was the observed early onset of change (*Figure 5A and B*). In looking to define the undefined state, we found that *Dillinger et al., 2017* previously reported that the undefined state is largely overlapping with both lamin-associated domains (LADs) and nucleolar-associated domains (NADs). In support of this observation, we too found that the undefined chromatin state is markedly gene poor and overwhelmingly overlapping with both empirically defined LADs and NADs (Z-score 90 and 150) that had previously reported in IMR-90 fibroblasts (*Figure 5—figure supplement 5Figure 5—figure supplement 4C*, *Figure 5—source data 5*; *Dillinger et al., 2017*; *Sadaie et al., 2013*). For reference, *Figure 5—figure supplement 4E* provides a chromosome level view of the significant overlap between replicative senescence accessible ATAC-seq peaks in the undefined state with NADs and LADs compared to all peaks annotated in WI-38 cells.

To control for the fact that the chromatin state annotations we used were derived from a different, albeit closely related cell line, we performed the same analysis using gene annotations which are cell line independent. We found that peaks in intergenic regions increase in accessibility with PDL on average which is consistent with the undefined (and gene poor) chromatin state annotation from IMR-90 cells. These results suggest that the chromatin states defined in IMR-90 cells are capturing the chromatin states in WI-38 cells (*Figure 5—figure supplement 4D*).

We tested and found that ATAC-seq peaks falling within NADs and LADs exhibit increased accessibility with replicative senescence (p < 10 e-16 for both: Wilcoxon rank-sum test). Notably, this trend was much greater in magnitude for peaks in NADs versus peaks in LADs, with a median log2 fold change of 0.98 for NADs vs. 0.24 for LADs (*Figure 5D*).

Digging deeper into NADs, we next asked the chromatin residing within NADs might respond differently than the rest of the genome to increasing PDL. We repeated the analysis shown in *Figure 5B* with two subsets; peaks excluded from NADs or peaks overlapping NADS (*Figure 5—figure supplement 5*). Dividing these two sets of peaks into the 25 chromatin states revealed that as expected, NAD excluded peaks mirror the pattern observed for significantly changing peaks across the whole genome in the original analysis. Conversely peaks overlapping NADs were found to increase in

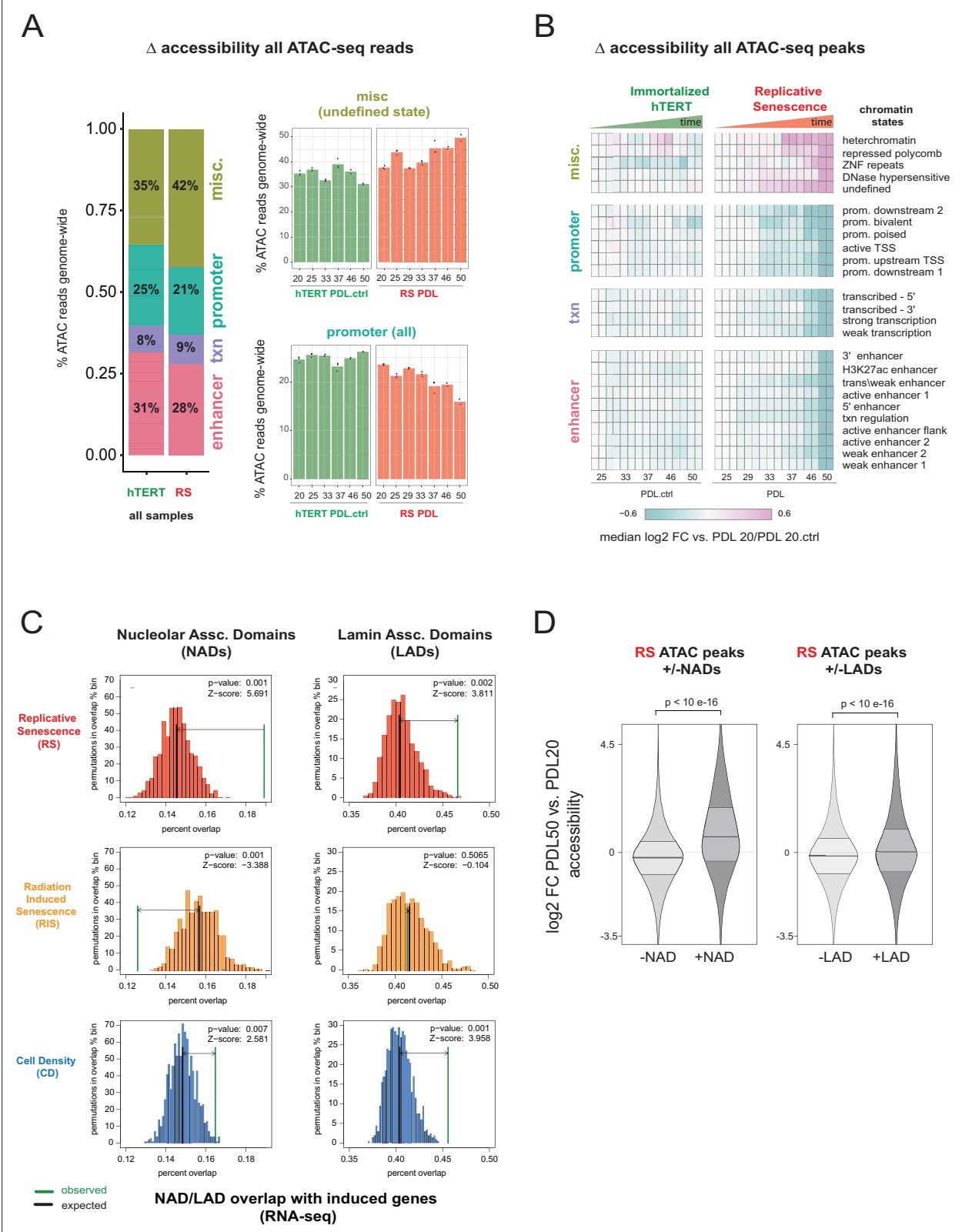

**Figure 5.** Increased accessibility within heterochromatin and nucleolar associated domains (NADs) is a dominant feature of the replicative senescence (RS) epigenome. (**A**) Percent of ATAC-seq reads falling into four broad chromatin states compiled from the ENCODE IMR-90 25 chromatin state prediction for all samples in WT or hTERT WI-38 cells. Percent of all ATAC-seq reads within two specific states (y-axis) vs PDL (x-axis) for WT (red) and hTERT (green) WI-38 cells (bar height denotes average of 3 replicates). (**B**) Median log2 fold change in ATAC-seq counts for significantly changing

*Figure 5 continued on next page*

*Figure 5 continued*

ATAC-seq peaks binned into ENCODE IMR-90 25 chromatin states (row) for each sample (column, n=2 or three replicates). Reference samples are PDL 20 or PDL 20.ctrl for WT and hTERT WI-38 cells respectively (not shown). (**C**) Observed and expected distribution of overlaps of significantly induced genes (from *Figure 1*, RNA-seq) with NADs and LADs. Replicative senescence (red), radiation-induced senescence (RIS) (orange), and cell density (CD) (blue). Median expected number of overlaps and observed number of overlaps shown by black and green lines respectively. (**D**) Average (n=2 or three replicates) log2 fold change distribution (PDL 50 vs. PDL 20) for all ATAC-seq peaks that overlap or are excluded from NADs (left) or LADs (right).

The online version of this article includes the following source data and figure supplement(s) for figure 5:

**Source data 1.** ENCODE IMR90 chromatin state labels.

**Source data 2.** ATAC-seq peak coordinates and metadata.

**Source data 3.** Normalized ATAC-seq coverage in peaks.

**Source data 4.** ATAC-seq fraction of reads in peaks.

**Source data 5.** IMR90 LADs from Salama 2013.

**Source data 6.** IMR90 NADs from Nemeth 2017.

**Figure supplement 1.** ATAC-seq library fragment distribution and size selection.

**Figure supplement 2.** ATAC-seq library fragment distribution after size selection and sequencing.

**Figure supplement 3.** ATAC-seq mitochondrial read percentages and ATAC-seq transcriptional start site enrichment.

**Figure supplement 4.** ATAC-seq QC metrics and extended analysis.

**Figure supplement 5.** Chromatin state profiles of ATAC-seq peaks in nucleolar-associated domains (NADs).

accessibility in a variety of chromatin states (e.g. poised promoters and many classes of enhancers) that exhibit reduced accessibility across the rest of the genome with increasing PDL.

One interpretation is that although the aggregate peak accessibility data appear to suggest that transcription is globally shutting down, the NAD specific view reveals rather that transcription is induced in a subset of loci. Although widespread, the reduced accessibility in euchromatin we observed in aggregate is not reflective of a pervasive shutdown of transcription across the entire genome. To formally test this hypothesis, we next asked if the shifts in NAD/LAD accessibility correlate with productive transcription and found that indeed, senescence induced genes significantly overlap NAD domains. Radiation and cell density induced genes exhibited weaker (cell density) or nonexistent (radiation) overlap with NADs and LADs. These data suggest that increasing accessibility in LADs and specifically NADs is a prominent feature of the replicative senescence epigenome that is directly connected to the senescent gene expression program (*Figure 5C*). Together these results highlight a striking increase in accessibility within nucleolar associated DNA that connects changes in the transcription with a global shift in the epigenome.

## Transcriptional regulators of the replicative senescence transcriptome and epigenome

To parse out the regulatory logic of the senescence gene expression program, we leveraged our ATAC-seq data to gain insight into which transcription factors regulate replicative senescence accessibility via transcription factor motif analysis. Having cataloged a universe of ATAC-seq peaks with significant changes in accessibility, we next assigned peaks to neighboring genes. Taking the top replicative senescence differentially expressed genes, we searched the proximal ATAC-seq peaks for enriched transcription factor motifs (*Figure 6A*). We found enriched motifs for TEAD1, CEBP family, SMAD family, AP1 complex, and FOX family transcription factors.

We also applied an orthogonal gene-independent methodology for determining which motifs are predictive of replicative senescence induced increases in ATAC-seq peak accessibility. Consistently, we again found TEAD1 motifs to be the most predictive feature (mean coefficient = 0.25 across 10 models-methods). In addition, we also found evidence for FOXE1 and SMAD1 regulation as well as other senescence related transcription factors, for example CEBPB and TP53 (*Figure 6B*).

Out of the FOX family transcription factors, *FOXE1* is unique in that its transcript exhibits one of the most specific (no change in hTERT, cell density, or radiation) and largest increases in replicative senescence gene expression (40–50 x) (*Figure 6—figure supplement 1*). FOXE1 is annotated as a thyroid-specific transcription factor with putative roles in thyroid development and cancer (*Zannini et al., 1997*; *Kallel et al., 2010*; *Ding et al., 2019*). It has also been reported to regulate two signaling

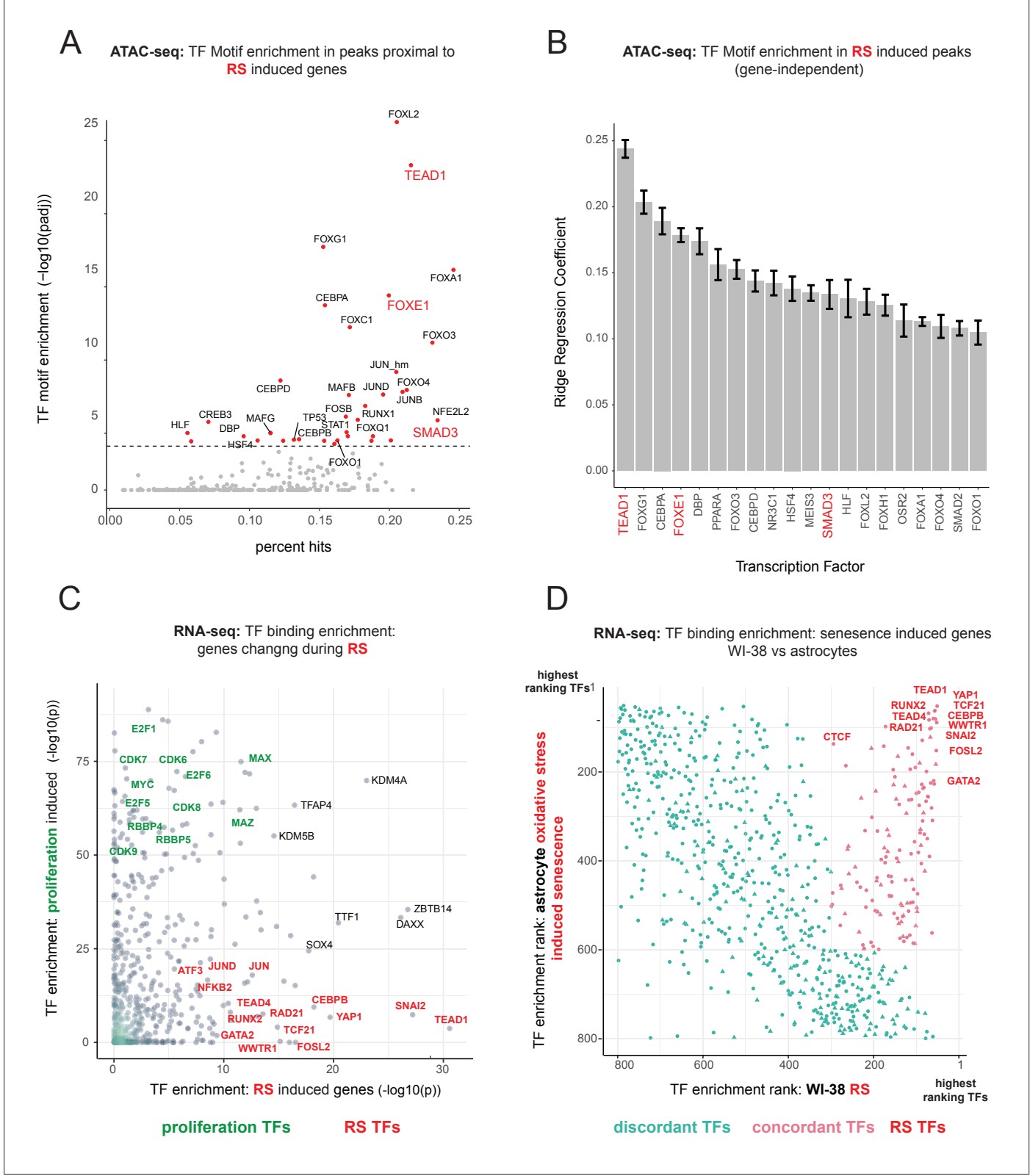

**Figure 6.** Master transcriptional regulators of replicative senescence. (**A**) Scatter plot of transcription factor motif enrichment in ATAC-seq peaks surrounding significantly induced genes (FDR adjusted p-value < 0.01, log2 FC >0.5) during replicative senescence. The Y-axis is the -log10 p-value; x-axis is the percent of input genes with the transcription factor motif. All points are transcription factors. Specific transcription factors of interest are called out in red. (**B**) Bar graph of ridge regression coefficient of motif predictive power in model of increasing peak accessibility with replicative

*Figure 6 continued on next page*

*Figure 6 continued*

senescence. Transcription factors of interest are highlighted in red. (**C**) Scatterplot of transcription factors enriched for binding in regulatory regions around replicative senescence depleted genes (y-axis, FDR adjusted p-value < 0.01, log2 FC <–0.5) vs. replicative senescence induced genes (x-axis, FDR adjusted p-value < 0.01, log2 FC <–0.5). Curated cell cycle transcription factors are colored in green; transcription factors of interest e.g. EMT/AP1/YAP1/TEAD1 etc. are colored in red. (**D**) Scatterplot of enriched transcription factors rank for binding enrichment in regulatory regions around senescence induced genes in astrocytes (y-axis) vs. replicative senescence induced genes in WI-38 cells (x-axis). transcription factors with discordant ranks/enrichment are colored in turquoise,transcription factors with concordant ranks/enrichment are colored in red; transcription factors of interest are labeled in red.

The online version of this article includes the following figure supplement(s) for figure 6:

**Figure supplement 1.** FOXE1 expression during senescence.

molecules upregulated in the replicative senescence time course; *TGFB3* and *WNT5A* (***Figure 1—source data 2***; ***Venza et al., 2011***; ***Pereira et al., 2015***). Furthermore, it has been reported that many of the FOX family transcription factors are relatively promiscuous binders of each others canonical motifs. Thus, it is possible that the increased accessibility in peaks with FOX motifs during replicative senescence may be driven by FOXE1 activity ***Rogers et al., 2019***. Despite the specificity and magnitude of replicative senescence induction, FOXE1's function in replicative senescence is unclear and warrants further investigation.

We returned to our bulk RNA-seq to test for enrichment of protein-DNA binding events (mapped by ENCODE) in regulatory elements proximal to our replicative senescence differentially expressed genes by using Landscape in silico deletion Analysis (LISA) (***Qin et al., 2020***). Plotting transcription factor enrichment for genes depleted with replicative senescence against genes induced during replicative senescence revealed three broad cohorts of transcription factors; proliferating cell transcription factors, replicative senescence transcription factors, and transcription factors whose binding was enriched around both sets of genes (***Figure 6C***). As expected, proliferation specific transcription factors are replete with cell-cycle-specific transcription factors for example E2F and RBP family transcription factors. A large portion of the transcription factors exhibiting replicative senescence specificity belong to four categories: inflammation transcription factors (NFKB, CEBPB), AP1 subunits (JUN,JUND,FOSL2), YAP1-TEAD1 components (TEAD1/4, YAP1, WWTR1), and EMT transcription factors (SNAI2, TCF21).

TEAD1 is a member of the TEA domain transcription factors whose functions range across a wide swath of biology depending on context and binding partner. TEAD transcription factors cannot induce gene expression without a cofactor, which is most often YAP1 (yes-associated protein 1) a key downstream effector of Hippo signaling (***Azakie et al., 2005***; ***Chen et al., 1994***; ***Benhaddou et al., 2012***; ***Yu et al., 2015***; ***Landin Malt et al., 2012***; ***Vassilev et al., 2001***; ***Zhao et al., 2008***; ***Ma et al., 2019***; ***Piccolo et al., 2014***). Consistent with our identification of TEAD1 and YAP1 as replicative senescence regulators, YAP1 activation has been tied to EMT, anti-apoptosis, telomere dysfunction, inflammation, and positive regulation of fatty acid oxidation (***Kurppa et al., 2020***; ***Lee et al., 2019***; ***Peng et al., 2017***; ***Zhang et al., 2017***; ***Chakravarti et al., 2020***).

Lastly, given that YAP1/TEAD1 activity appears to increase during replicative senescence against the backdrop of an altered epigenomic context, we tested for an interaction between TEAD1 motifs and the increasingly accessible NADs and LADs. We found that not only are TEAD1 sites significantly enriched within NADs and LADs (p < 7.17e-07, and p < 6.12e-03, respectively hypergeometric), but we also discovered a greater-than-additive increase in accessibility with replicative senescence for TEAD1 motifs that occur within a NAD (p < 3.12 e-07). These results suggest that there is a functional connection between TEAD1 activity and NAD domains during replicative senescence. Thus, The changing epigenetic context of replicative senescence may alter TEAD1 binding and target expression induction with increasing PDL.

Collectively, these analyses uncover a common theme amongst putative regulatory transcription factors; Hippo signaling (YAP1/TEAD1), EMT transcription factors, and TGF-β signaling (SNAI2, SMAD activity). SNAI2 has been shown to work in tandem with the YAP1/TEAD1 complex and these pathways often work towards similar biological ends (***Tang et al., 2016***; ***Kurppa et al., 2020***). Together, these transcription factors are reported as highly involved with proliferation, EMT, ECM production, fibrosis, and apoptosis avoidance (***Kim et al., 2019a***).

Lastly, given that the LISA results are based on binding events collected from a vast multitude of cell lines, we wondered if the same transcription factors might be found to regulate senescence in a completely different cellular context. To test this, we took significantly induced genes from a senescence model using astrocytes and oxidative stress as the senescence trigger (*Crowe et al., 2016*). Plotting transcription factor enrichment for genes induced in astrocyte senescence against transcription factor enrichment for WI-38 replicative senescence genes revealed that although there were a substantial number of discordant transcription factors, there was a clear population of transcription factors highly enriched in both senescence models (*Figure 6D*). Notably, the top concordant transcription factors ranked at the top in both contexts and recapitulate all the previous results for example YAP1/TEAD1, SNAI2, CEBP family transcription factors and AP1 subunits.

## scRNA-Seq trajectory analysis resolves WI-38 cells' approach to replicative senescence

A wealth of recent work in single-cell transcriptomics has demonstrated that ordering single cells in a process-specific trajectory often reveals nuanced timing and dynamics of gene expression that bulk assays cannot capture (*Qiu et al., 2017*; *Trapnell et al., 2014*). Mapping this trajectory is frequently referred to as 'pseudotime analysis'. We employed pseudotime analysis to arrange single WI-38 cells along a pathway to senescence setting proliferating cells as the trajectory 'root'. As expected, early and late PDL cells concentrated at the beginning and end of the pseudotime trajectory respectively (*Figure 7A*). We next performed differential expression analysis to identify genes that change significantly over pseudotime. We plotted examples of genes changing early (*CENPK*–an S-phase cell-cycle-regulated gene), midway (*SNAI2*–a master regulator of EMT), or late (*PAPPA*–a prominent SASP factor) in pseudotime in *Figure 7B*.

We next generated gene expression trajectories for the top (by significance p-value < 0.001) 5000 differentially expressed genes across pseudotime (*Figure 7—figure supplement 1*, *Figure 7—source data 1*). To identify the temporal relationship between the biological processes and transcription factors that compose replicative senescence in WI-38 cells, we clustered and ordered pseudotime trajectories (*Figure 7—figure supplement 2*). Broadly, we classified the pseudotime expression pattern as early, transition, or late based on the maximum median value of all constituent genes for each cluster (*Figure 7C*). We performed LISA transcription factor and GO enrichment analysis on each cluster and used our findings to assign putative functional labels across pseudotime (*Figure 7D*, *Figure 7—figure supplement 2*, *Figure 7—source data 1* and *Figure 7—source data 2*).

Early pseudotime is dominated by the transcription factors (E2F) and GO terms associated with cell cycle progression through the G2M and S phases. Moving down the y-axis deeper into pseudotime, we observe that the next primary cluster of transcription factors and functional annotations exhibited widespread enrichment across all of pseudotime. Furthermore, transcription factors enriched in this cluster are involved with basic cellular functions (e.g. euchromatin maintenance, transcription, growth) and are likely representative of normal WI-38 function in G1 phase.

Moving forward into the transition phase of pseudotime, we observed an enrichment of transcription factors regulating higher order chromatin structure (CTCF) and epigenetic silencing (polycomb group complex). Also in the transition region of pseudotime, the earliest enrichments appear for transcription factors related to EMT (SNAI2), TGF-β signaling (SMAD3), YAP1/TEAD1 activity, and the AP1 complex (JUN, FOS, FOSL2). We observed continued enrichment for these transcription factors and processes throughout the rest of the transition phase and into late pseudotime.

Lastly, late in pseudotime, we observed enrichment of transcription factors and functional annotations related to regulation of inflammatory processes (NFKB1, RELA, ERG1, CEBPB) and changes in cellular morphology (*Figure 7D and Figure 7—figure supplement 2*). These observations are consistent with observations made with bulk RNA-seq that replicative senescence WI-38 cells exhibit transcriptomic features similar to that observed in TGF-β signaling, EMT, and with YAP1/TEAD1 activity. Collectively, these results present a possible order of operations for replicative senescence progression that highlights an initial cessation of active mitotic cycling, followed by an epigenetic shift that precedes a strong EMT/TGF-β signal before segueing into a pro-inflammatory secretory state.

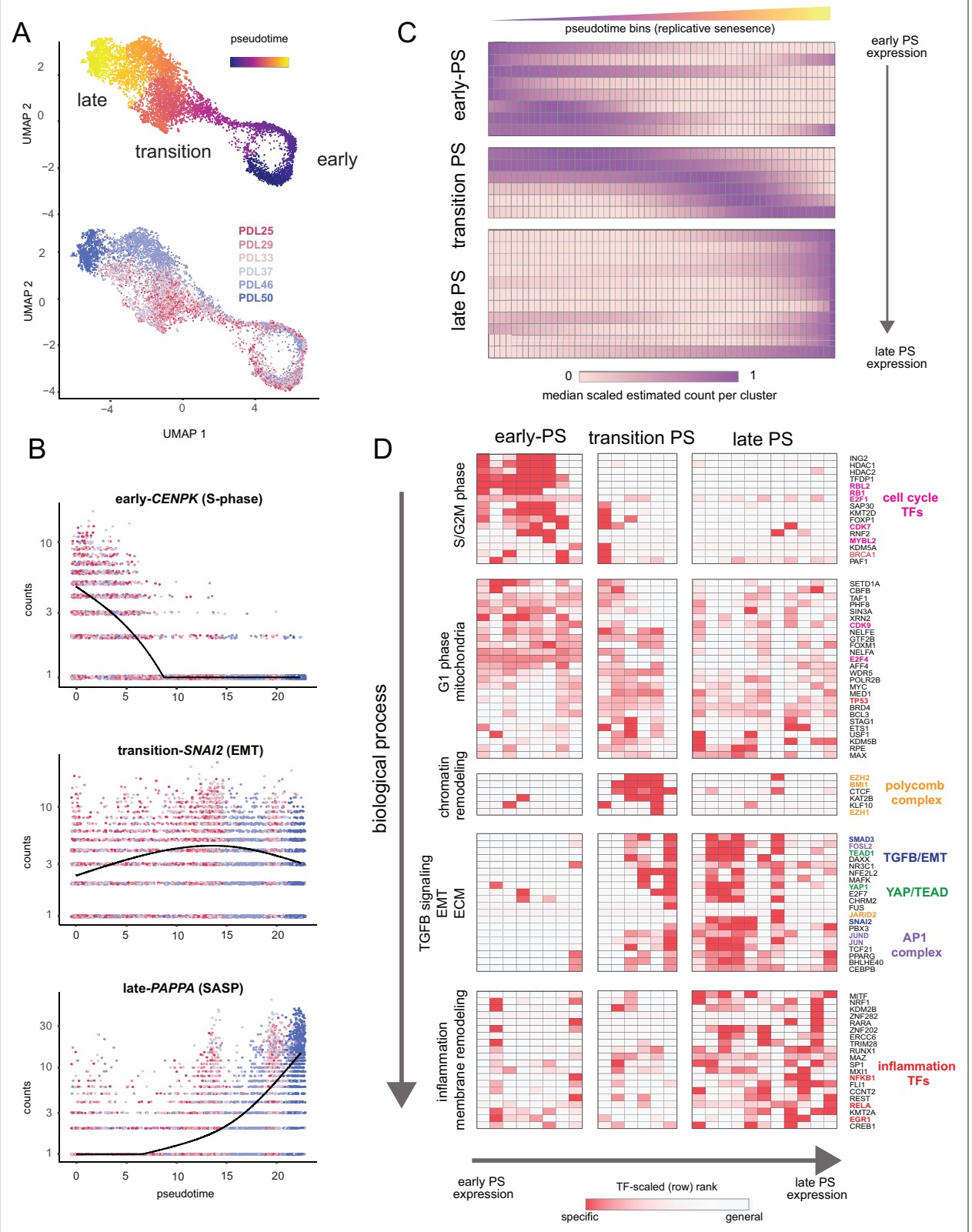

**Figure 7.** Pseudotime (PS) analysis of WI-38 approach to replicative senescence using single-cell RNA-seq. (**A**) UMAP projection of single WI-38 cells collected at increasing PDLs (PDL25–red) to (PDL50–blue) and colored by pseudotime (top). (**B**) Scatterplot of single-cell gene expression across pseudotime with three genes representative of changes that occur early in pseudotime (top), during the transition phase (middle) and in late pseudotime (bottom). Each point is a cell colored by PDL–PDL25 (red) to PDL50 (blue). x-axis is pseudotime, and y-axis is counts for the gene of interest.

*Figure 7 continued on next page*

*Figure 7 continued*

Black line is a cubic spline. (**C**) Hierarchical clustering and heatmap of smoothened gene expression trajectories over 60 pseudotime bins (x-axis) of 25 K-median gene expression clusters (y-axis). Value plotted is the scaled (min expression to max expression; 0–1) median expression of all genes in the cluster. Clusters are divided into three (early, transition, late) pseudotime categories. (**D**) LISA transcription factor enrichment analysis using the 25 clusters from A divided into the same three (early, transition, late) pseudotime categories. Transcription factors and clusters are further divided on vertical axis into putative groupings based on transcription factor functions and GO term enrichment.

The online version of this article includes the following source data and figure supplement(s) for figure 7:

**Source data 1.** Gene expression changes with pseudotime.

**Source data 2.** GO term enrichment in pseudotime clusters.

**Figure supplement 1.** Heatmap of genes changing with senescence pseudotime.

**Figure supplement 2.** Heatmap of enriched gene sets for replicative senescence pseudotime clusters.

## Senescent WI-38 fibroblasts express canonical myofibroblastmarkers and metabolic features

Given the repeated observations linking replicative senescence with EMT, (*Figure 1*, *Figure 3*), TGF-β and YAP1/TEAD1 activity (*Figure 6*, *Figure 7*), we considered the possibility that these processes are connected through the fibroblast to myofibroblast transition (FMT), a subtype of EMT in which stressed or injured fibroblasts differentiate into myofibroblasts (*Phan, 2008*; *Piersma et al., 2015*; *Hinz, 2007*). Upon receiving cues mediated by injury or stress (e.g. activated TGF-β), fibroblasts can trans-differentiate into myofibroblasts, whose functions as 'professional repair cells' include increased proliferation, migration, apoptosis avoidance, cell and tissue contraction, and ECM/collagen deposition to promote tissue repair and wound closure (*Hinz and Lagares, 2020*; *Gibb et al., 2020*).

Previous work has demonstrated that there exists mechanistic and functional association between telomerase inhibition, senescence, and myofibroblasts. Senescence is an integral part of the wound healing processes; upon injury resolution, activation of a senescence-like phenotype prevents unchecked collagen secretion and fibrosis by preventing myofibroblast proliferation and earmarking them for subsequent immune clearance (*Demaria et al., 2014*; *Krizhanovsky et al., 2008*; *Jun and Lau, 2010*; *Mellone et al., 2016*; *Razdan et al., 2018*; *Liu et al., 2002*; *Liu et al., 2006*).

To further explore this proposition, we retrieved canonical myofibroblast marker genes and direct transcriptional targets of the YAP1/TEAD1 complex to determine to what extent these genes are expressed in WI-38 replicative senescence cells at both the RNA and protein level. First, we examined canonical myofibroblast markers (*Hinz and Lagares, 2020*), collagen produced by myofibroblasts (*Zhang et al., 1994*), genes upregulated in myofibroblasts derived from idiopathic pulmonary fibrosis (IPF) patients (*Strunz et al., 2020*), and effectors and targets of TGF-β signaling in two data modalities: bulk RNA-seq and bulk proteomics (*Figure 8A*).

For the majority of genes in the curated myofibroblast panel, expression of transcripts and protein increased with PDL. Importantly, expression of smooth-muscle actin (ACTA2), a classic myofibroblast marker, increases strongly in replicative senescence. The expression of follistatin-like protein (FSTL1), also known to be strongly expressed in smooth muscle, shows a similar pattern, as does fibrillin (FBN1). All three of these are associated with smooth muscle and TGF-β family regulation. In addition we observed an increase in both fibrillar and basal lamina collagens in our data at both the RNA and protein levels.

Collagen processing is a multi-step process requiring the coordination of multiple enzymes and metabolites (*Lodish and Zipursky, 2001*). Review of the collagen synthesis pathway alongside our metabolomic and proteomic data provide further confirmation that WI-38 replicative senescence cells exhibit altered collagen metabolism (*Figure 8B*). We observed upregulation of multiple pathway enzymes as well as increased abundance of hydroxyproline, a primary constituent amino acid of collagen protein. In addition, we observe a striking depletion of ascorbate (vitamin C) which is required for proline hydroxylation and is an essential vitamin. It is possible that our observations underestimate the collagen production potential of replicative senescent WI-38 cells as they appear limited in terms of collagen production by the amount of supplemented vitamin C (*Boyera et al., 1998*).

*Mellone et al., 2016* recently reported that although senescent fibroblasts share features with myofibroblasts, this resemblance does not extend to fibrogenic ECM components, e.g. collagen (*Mellone et al., 2016*). However, it is important to note that *Mellone et al., 2016* focused on radiation

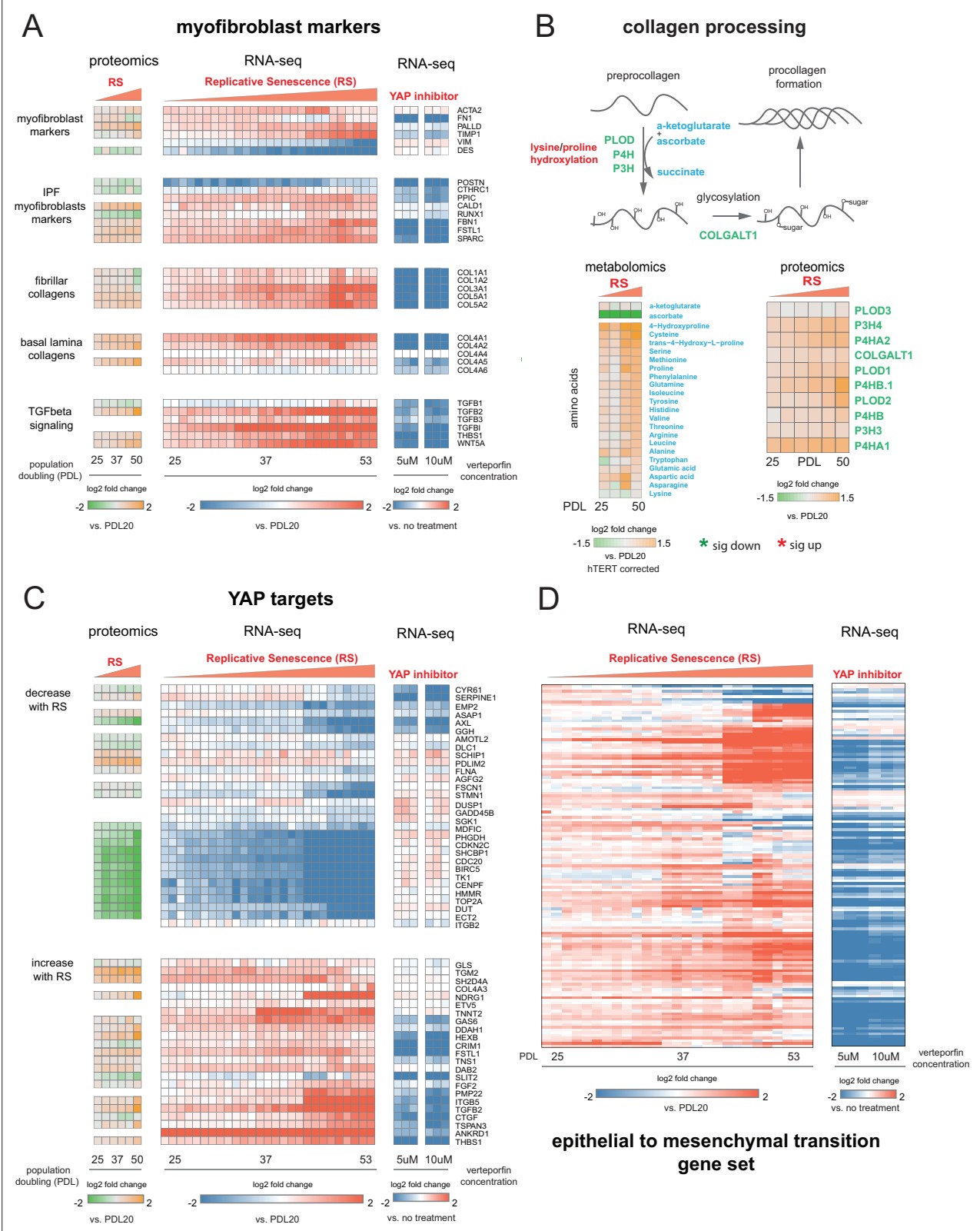

**Figure 8.** YAP regulation of myofibroblast markers and YAP1/TEAD1 targets during replicative senescence. (**A**) RNA-seq and proteomics heatmaps of selected genes based on myofibroblast markers and biology. Values plotted are log2 fold change of each PDL vs. PDL 20 for replicative senescence timecourse and vs. no treatment for late PDL (PDL 40) WI-38 cells treated with the YAP inhibitor verteporfin. Proteomics values are median values for n = 3 replicates. Values for individual RNA-seq replicates (n = 3) are shown. (**B**) Diagram of collagen processing (left) with metabolites in blue and proteins

*Figure 8 continued on next page*

*Figure 8 continued*

in green. Collagen specific amino acid derivatives are outlined in red. Heatmap of log2 fold changes for metabolites and proteins involved in collagen processing (right). Proteomics values are median values for n = 3 replicates. Metabolomics values are hTERT batch corrected median values for n = 4 replicates. (**C**) Heatmaps of YAP1 targets collected by *Kurppa et al., 2020.* for proteomics and RNA-seq during replicative senescence and in late PDL (PDL 40) WI-38 cells treated with the YAP inhibitor verteporfin. Values plotted are same as in A. (**D**) Heatmap of log2 fold changes relative to PDL 20 or untreated control of leading edge genes from gene set enrichment analysis driving the hallmark EMT signature in verteporfin-treated WI-38 cells and during replicative senescence. Values for individual RNA-seq replicates (n=3) are shown.

The online version of this article includes the following source data and figure supplement(s) for figure 8:

**Source data 1.** RNA-seq data for verteporfin treatment.

**Figure supplement 1.** Effect of inhibiting the YAP1/TEAD1 interaction with verteporfin treatment on WI-38 cells.

---

induced senescence rather than replicative senescence, and we similarly observed less or no induction of many of these same genes in our radiation induced senescence experiment. In addition, it is possible that with more time ( > 10 days), fibrogenic ECM expression could initiate in the irradiated WI-38 cells.

Moving forward in our myofibroblast panel, we observed that expression of TGF-β cytokine, TGF-β1, decreased significantly with replicative senescence in both RNA and protein. However, we see robust induction of the TGF-β isotype 2 (TGF-β2) cytokine with replicative senescence (*Figure 8A*) as TGF-β1 abundance drops, which indicates a switch in TGF-β isotypes with replicative senescence. It has been shown TGF-β2 is a more potent inducer of the endothelial to mesenchymal transition (EndMT) in vitro compared to TGF-β1 and TGF-β3 in human microvascular endothelial cells, and TGF-β2 may be playing a similar role here in inducing FMT and replicative senescence in WI-38 cells (*Sabbineni et al., 2018*). The distinct functional roles of different TGF-β isotypes are largely unknown, although both are known to activate the SMAD transcription factors. On the basis of our data, it seems likely that the TGF-β paralog relevant here might not be TGF-β1, but TGF-β2.

## Expression of YAP1/TEAD1 targets during replicative senescence

Next, we retrieved a gene set of YAP1/TEAD1 targets assembled by Kurrpa et al. from five separate studies (*Kurppa et al., 2020*). Taking the intersection of the five YAP1/TEAD1 gene target lists, we only kept genes present in at least two of the studies and plotted the remainder in *Figure 8C* across the two data types. We arranged the YAP1/TEAD1 targets by the two predominant expression patterns–decrease with replicative senescence (top) and increase with replicative senescence (bottom). As with myofibroblast markers, the data from both modalities is largely concordant.

Interestingly, we noted a striking bifurcation whereby YAP1 targets tend toward either strong down-regulation or strong up-regulation. The down regulated partition is heavily enriched for classic cell cycle regulators such as TOP2A, CDC20, BIRC5, and CDK9 suggesting that this dichotomy in YAP1 activity is heavily influenced by the cell cycle and consistent with recent work (*Kim et al., 2019b*; *de Sousa et al., 2018*).

Excitingly, TGF-β2 was one of the two YAP1 targets found in four out of five collected YAP1/TEAD1 data sets, supporting the inference that the TGF-β2 and not TGF-β1 is the relevant paralog and potentially regulated by YAP1/TEAD1 in the replicative senescence context. In addition, another YAP1 target, thrombospondin-1 (THBS1), is also upregulated at both the protein and transcript level with RS; THBS1 is known to act as an activator of TGF-β signaling, and specifically TGF-β2, through proteolytic cleavage of latent TGF-β (*Ribeiro et al., 1999*).

## Inhibiting the YAP1/TEAD1 interaction in WI-38 cells suppresses expression of YAP1/TEAD1 targets and the replicative senescence gene signature

In all, the data reveal that replicative senescence WI-38 cells share multiple defining transcriptomic and proteomic features with myofibroblasts. Furthermore, we found that a subset of YAP1/TEAD1 targets are both induced with replicative senescence, and are principal components of TGF-β signaling (TGF-β2, THBS1). Thus, the YAP1/TEAD1 complex may be acting in convergence with TGF-β signaling with increasing PDL to enact a myofibroblast-like state that we recognize as replicative senescence.

Recently, *Mascharak et al., 2021* showed verteporfin treatment of wounds in mice alleviated the fibrotic state during wound healing, reverted the profibrotic transcriptional program, and reduced the myofibroblast population. Verteporfin is also known to be an inhibitor to the formation of the YAP1/TEAD1 complex (*Wang et al., 2016*). To test for a direct role for YAP1/TEAD1 in transcriptional regulation of YAP target genes, myofibroblast marker expression and the EMT signature during replicative senescence transition, we treated late passage WI-38 PDL 40 cells with verteporfin or DMSO as an untreated control. We utilized RNA-seq to capture the gene expression changes on cells treated with verteporfin. First, to confirm the verteporfin treatment inhibited expression of canonical YAP target genes in WI-38 cells, we checked the expression of *CTGF*, *CYR61*, and *TGFB2* transcripts (*Figure 8—figure supplement 1*; *Zhao et al., 2008*; zhang2009tead; chen2001angiogenic). We found that treatment with 10 µM verteporfin for 2 hours reduced expression of these three YAP target genes by 70% relative to the no treatment control (*Figure 8—figure supplement 1*, *Figure 8—source data 1*). We expanded the analysis to the myofibroblast markers and YAP targets used in *Figure 8* and observed that verteporfin treatment reduces the gene expression of most myofibroblast markers and collagen and many YAP1 target genes that were shown to increase with replicative senescence (*Figure 8A and C*).

We applied GSEA analysis using the MSigDB Hallmark annotation to understand what pathways are affected by verteporfin treatment of cells using the RNA-seq data from the verteporfin treated samples relative to non-treated control. Consistent with our hypothesis that YAP1/TEAD1 plays a role in regulating the FMT-like transition of cells undergoing replicative senescence, treatment of late passage cells with verteporfin resulted in negative enrichment of pathways relevant to FMT and TGF-β signaling (*Figure 8—figure supplement 1B*). We also took genes driving the EMT annotation enrichment during replicative senescence to see if inhibition of YAP1/TEAD1 complex formation aids in reducing EMT pathway gene expression. Compared to the no treatment control, the majority of genes induced during replicative senescence that drive the EMT enrichment exhibit a striking decrease in expression after two hours of treatment. These data suggest that the use of verteporfin to inhibit the YAP1/TEAD1 interaction can impede the FMT-like transition observed in replicative senescent cells (*Figure 8—figure supplement 1D*).

To gain a global understanding of how inhibiting the YAP1/TEAD1 interaction affects the transcriptomic landscape of late passage WI-38 cells, we compared the gene expression signature of replicative senescence cells to that of verteporfin-treated cells and found that verteporfin treatment globally reverses many of the gene expression changes occurring with replicative senescence (*Figure 8—figure supplement 1*). This results suggest that YAP1/TEAD1 transcriptional activity is a crucial regulator of a large swath of the replicative senescence transcriptome.

## Discussion

The study of replicative senescence in human tissue culture has proven to be an informative model for learning how genetic and environmental factors impact cellular senescence. However, the field has not fully taken advantage of the 'omics' revolution. Rekindled by the advent of senolytics, interest in the molecular underpinnings of senescence has burgeoned in recent years as researchers seek to design therapeutic strategies for ablating senescent cells (*Amor et al., 2020*; *Elmore et al., 2018*; *Wagner and Gil, 2020*; *Aghajanian et al., 2019*). However, many such studies span only one or two systematic data modalities. Here, to fully leverage the power of recent advances in high-dimensional profiling, we revisit the original Hayflick limit in WI-38 lung fibroblasts cells with a battery of assays including RNA-seq, ATAC-seq, scRNA-seq, proteomics, and metabolomics in an effort to capture the defining features of replicative senescence at every step of the central dogma and beyond. Our results are summarized graphically in *Figure 9*.

One important feature of our study is the number and type of control conditions. From our pilot studies, it was clear that studying replicative senescence in isolation would preclude our ability to know what features of change were specific to replicative senescence versus high cell density and/or DNA-damage-induced growth arrest. Thus, we included cellular density as an alternate method for arresting growth and exposure to ionizing radiation as an alternate methodology for senescence induction and DNA damage response. Lastly, all of our modalities were paired with samples collected from an hTERT immortalized cell line grown in parallel. Importantly, very little changed across all modalities in the hTERT cell line consistent with the changes we highlight in nine as specific to replicative senescence.

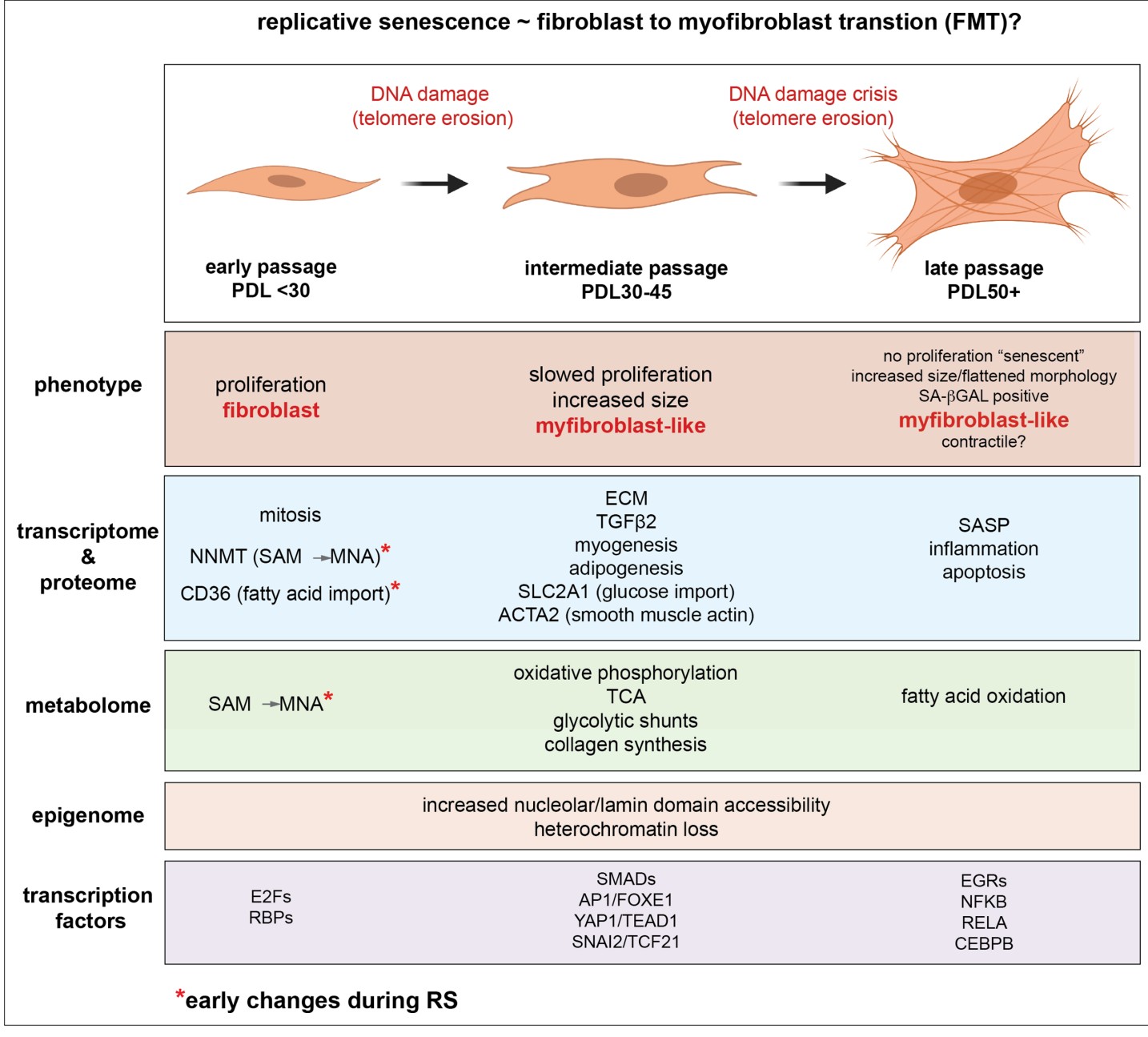

**Figure 9.** Replicative senescence fibroblast to myofibroblast transition (FMT) model. We divided Replicative senescence progression into three major categories and summarized our results across all data modalities focusing on features in common with myofibroblasts.

The kinetics and precise timing of senescence onset have been obscured by low temporal resolution and ensemble measurements that cannot differentiate between global shifts in gene expression versus changing proportions of senescent cells (*Passos et al., 2007*; *Xu et al., 2013*; *Victorelli and Passos, 2017*; *Smith and Whitney, 1980*; *Tang et al., 2019*; *Wiley et al., 2017*; *Nassrally et al., 2019*; *Smith and Whitney, 1980*). Here, with a combination of high time resolution and single-cell RNA-seq, we provide evidence that the early manifestation of the senescent gene expression reflects gradual changes on a per cell basis rather than changing cell proportions. In effect, individual cells 'show their age'" with increasing PDL long before permanently exiting the cell cycle and transiting fully into the senescent state. The implications of this conclusion extend to organismal aging. For example, the percentage of senescent cells calculated from aging organisms varies greatly depending on the marker/phenotype used (*Ogrodnik, 2021*). The reported disparities could be explained in

part by the use of early versus end stage markers. Likewise, it is possible the reported increase in fibroblast heterogeneity and altered functionality with age is a direct result of cells slowly moving along a spectrum towards replicative senescence (*Shin et al., 2020*; *Mahmoudi et al., 2019*). Importantly, the gradual progression suggests that cells need not reach the endpoint to elicit a phenotype. For instance, proliferative fibroblasts isolated from IPF patients exhibited multiple senescent features and phenotypes in addition to accelerated senescence progression (*Yanai et al., 2015*). Lastly, this phenomenon is not constrained to fibroblasts as we observe that the salient regulatory features of replicative senescence extend to cell types as distant as astrocytes (*Figure 6D*).

In our data, the pattern of gene expression annotated to EMT as a unique feature of replicative senescence that consistently presents early and robustly at both the RNA, protein, and single cell level. Given the fundamental nature of the EMT transition with respect to cellular function (development, fibrosis, and wound healing), it is not surprising that this hallmark tracks with multiple proteins and gene sets suggesting drastic metabolic rewiring. In our metabolic data, we highlight shifts in carbon and fatty acid utilization that have been reported previously as hallmark metabolic features of EMT. These metabolic changes demonstrate that our observations represent an authentic change in cellular state as opposed to a superficial uptick in a few EMT-related genes.

The data presented above across multiple data modalities to provide a clear connection between senescent cells and myofibroblasts supported by independent observations at the level of DNA, RNA, protein, transcription factor activity and metabolism (*Figure 9*). In light of these, replicative senescence resembles a specialized subtype of EMT specific to the trans-differentiation of fibroblasts into myofibroblasts (FMT) in response to wound healing (*Lombardi et al., 2019*; *Gibb et al., 2020*; *Hinz and Lagares, 2020*). We hypothesize that during fibrotic disease states and/or age, fibroblasts migrate to sites of micro-injuries. As these proliferating fibroblasts become replicatively aged, they are triggered (by DNA damage or other insults) to rewire their metabolism to induce FMT via active epigenetic reorganization (NNMT-SAM/NAD sink). It is important to note here that the in vitro DNA damage here arises primarily from telomere erosion documented by the observation that our hTERT control cultures do not exhibit the same changes. However, genotoxic stress in vivo may originate from a variety of endogenous and environmental sources e.g. reactive oxygen species, replication stress, chemical exposure etc.

Following increases in DNA accessiblity, expression of newly opened TEAD1/YAP1/SMAD target genes cement FMT transition by promoting fibrosis and a myofibroblast-like, ECM-secreting state. This model is supported by the synergy between TEAD1 motifs and NAD domains we report and reconciles conflicting reports that YAP1/TEAD1 inhibition can both prevent and promote senescence (*Fu et al., 2019*; *Kurppa et al., 2020*; *Xie et al., 2013*; *Jia et al., 2018*). Basically, the functional consequences of YAP1/TEAD1 inhibition will depend on the epigenetic organization of the cells used.

Further metabolic changes (hexoasmine/collagen synthesis and fatty acid oxidation) then support the new pro-fibrotic state. Finally, end point senescent cells reinforce the senescent state and contribute to neighboring cell progression towards senescence via secretion of inflammatory factors and SASP.

We think that this transition is distinct from classic FMT as the endpoint cells are not proliferative, but instead bear striking resemblance to lingering senescent myofibroblasts that can persist long after wound repair is complete (*Hinz and Lagares, 2020*). Rather than a privileged or unique state, perhaps replicative senescence is better categorized as a DNA damage mediated path to a potentially common stress-induced endpoint.

The complex interplay between metabolism and epigenetic regulation preclude easy determination of a causal factor in translating DNA damage into the replicative senescence/EMT program in WI-38 cells. Does EMT regulate metabolism or vice versa? Here we present compelling evidence on the side of metabolism. We observe both an early and sharp rise in NNMT expression and activity in addition to a global increase in heterochromatin accessibility. These results are consistent with NNMT's reported role as global epigenetic regulator through its methylation sink activity (*Komatsu et al., 2018*; *Eckert et al., 2019*; *Pissios, 2017*; *Ulanovskaya et al., 2013*).

Importantly, the observed shifts in repressed chromatin induced by NNMT are functional and may play a central role in fibroblast biology and stress response in multiple contexts. First, *Eckert et al., 2019* recently demonstrated that NNMT activity and the resulting heterochromatin reorganization initiate the expression program of cancer-associated fibroblasts (CAFs) associated with oncogenic

stroma in vivo (*Eckert et al., 2019*). Similar to senescent fibroblasts and myofibroblasts, the defining features of CAFs are increased cytokine production, metabolic rewiring, and ECM alteration and production (*Sahai et al., 2020*). Second, NNMT is one the most upregulated genes in a TGF-β-mediated in vitro FMT conversion in WI-38 cells (*Walker et al., 2019*). Finally, the changes in silenced chromatin we observed during replicative senescence overlap with induction of gene expression driving the senescent phenotype (*Figure 5D*). In fact, pseudotime analysis argues that chromatin reorganization may precedes the FMT induction as evidenced by the loss of polycomb activity that appears prior to enrichment of the EMT and YAP1/TEAD1 transcription factors enrichments (*Figure 7D*).

The path to replicative senescence process has many in vivo parallels with implications for aging and pathogenesis. After observing induction in WI-38 cells of IPF myofibroblast markers (*Figure 8A*), we expanded our literature search and found a striking overlap (TGF-β signaling, YAP1 activity, and EMT) between our data and scRNA-seq expression profiles from alveolar epithelial cells collected from IPF patients (*Xu et al., 2016*). These studies also report a large induction of TGF-β2 relative to TGF-β1 which is consistent with our findings and highlights a clear connection between in vitro replicative senescence and an in vivo disease state.

Another example of such a connection arises from our LISA analysis: one of the top enriched transcription factors, TCF21, has been implicated in atherosclerotic disease progression. *Wirka et al., 2019* found that TCF21 promotes the transition of vascular smooth muscle cells into a novel fibroblast-like cell type they dub 'fibromyocytes' owing to their possession of both fibroblast and myocyte phenotypes in atherosclerotic lesions in both mice and humans.

In general, the FMT hypothesis provides a conceptual framework and integrative model for linking the multi-modal senescent phenotypes we observed to multiple human age-related diseases. Given the observation that fibrosis and senescence markers correlate with increasing age in multiple tissues, it is possible that FMT might be a widespread phenomenon underlying many age-related pathologies (*Yousefzadeh et al., 2020*; *Idda et al., 2020*). Future work harnessing multi-modal single cell technology coupled with relevant in vivo models will aid greatly in determining the exact order of events and physiological import.

# Materials and methods

## Key resources table

| Reagent type (species) or resource | Designation | Source or reference | Identifiers | Additional information |
|---|---|---|---|---|
| Cell Line (*H. sapiens*) | WI-38 fibroblasts | Coriell | AG06814-N | |
| Cell Line (*H. sapiens*) | WI-38 hTERT | This paper | | immortalized WI-38 human fibroblasts, methods |
| Transfected construct (*H. sapiens*) | pCDH-CMV-hTERT-EF1a-puro | This paper | | Lentiviral plasmid, methods |
| Commercial assay, kit | Senescence-β-Galactosidase | Cell Signaling Tech. | 9,860 | |
| Commercial assay, kit | MycoAlert Mycoplasma Detection | Lonza | LT07-218 | |
| Commercial assay, kit | Direct-zol RNA Miniprep Plus | Zymo Research | R2072 | |
| Commercial assay, kit | Chromium Single Cell 3' v2 | 10 x Genomics | 120,237 | |
| Commercial assay, kit | Chromium Single Cell A Chip | 10 x Genomics | 1000009 | |
| Commercial assay, kit | Tagment DNA Enzyme and Buffer | illumina | 20034197 | |
| Commercial assay, kit | Clean-and-Concentrator-5 | Zymo Research | D4014 | |
| Commercial assay, kit | NEBNext High-Fidelity 2 X PCR | NEB | M0541L | |
| Commercial assay, kit | TruSeq Stranded mRNA Library | illumina | 20020595 | |
| Commercial assay | Bioanalyzer High Sensitivity DNA | Agilent | 5067–4626 | |
| Commercial assay, kit | Pierce BCA Protein Assay | Thermo Fisher | 23,227 | |

*Continued on next page*

*Continued*

| Reagent type (species) or resource | Designation | Source or reference | Identifiers | Additional information |
|---|---|---|---|---|
| antibody | anti-human p16 antibody (Mouse monoclonal) | BD Biosciences | RRID:AB_395229 | (1:250) |
| antibody | anti-human p21 antibody (Mouse monoclonal) | BD Biosciences | RRID:AB_396414 | (1:250) |
| Commercial assay, kit | Pippin Prep 2% 100–600 bp | Sage Science | CDF2010 | |
| Chemical compound, drug | Verteporfin | RD Systems | 1243926 | |
| Other | Zorbax Extend C18 column | Aglient | 759700–902 | |
| Other | SeQuant ZIC-pHILIC column | EMD Millipore | 150,460 | |
| Software, algorithm | R (v4.0.3 and 3.6.2) | r-project.org/ | RRID:SCR_001905 | |
| Software, algorithm | Salmon (v 0.8.2) | combine-lab.github.io/salmon/ | RRID:SCR_017036 | |
| Software, algorithm | DESeq2 (v1.30.1) | bioconductor | RRID:SCR_015687 | |
| Software, algorithm | sva package(v3.38.0) | bioconductor | RRID:SCR_012836 | |
| Software, algorithm | fgsea 1.16.0 | bioconductor | RRID:SCR_020938 | |
| Software, algorithm | CellRanger 3.0 | 10 x Genomics | RRID:SCR_017344 | |
| Software, algorithm | SCTransform (v 0.3.2) | Satijalab.org/seurat | | |
| Software, algorithm | Seurat (v4.0.1.9005) | satijalab.org/seurat | RRID:SCR_007322 | |
| Software, algorithm | monocle3 (v1.0.0) | cole-trapnell-lab.github.io/monocle3 | RRID:SCR_018685 | |
| Software, algorithm | bowtie2 (v2.3.4.1) | bowtie-bio.sourceforge.netbowtie-bio.sourceforge.net | RRID:SCR_016368 | |
| Software, algorithm | samtools (v1.2) | http://ww.htslib.org/ | RRID:SCR_002105 | |
| Software, algorithm | Picard (v2.6.4) | broadinstitute.github.io/picard | RRID:SCR_006525 | |
| Software, algorithm | macs2 (v2 2.1.2) | hbctraining.github.io/main/ | RRID:SCR_013291 | |
| Software, algorithm | GenomicRanges (v1.42.0) | bioconductor | RRID:SCR_000025 | |
| Software, algorithm | cutadapt (v2.4) | https://github.com/marcelm/cutadapt | RRID:SCR_011841 | |
| Software, algorithm | bcl2fastq (v2.20) | Illumina | RRID:SCR_015058 | |
| Software, algorithm | regioneR v1.22.0 | bioconductor | | |
| Software, algorithm | ATACseqQC v1.14.4 | bioconductor | | |
| Software, algorithm | LIMMA v3.46.0 | bioconductor | RRID:SCR_010943 | |
| Software, algorithm | Qvalue v2.26.0 | combine-lab.github.io/salmonQvalue | RRID:SCR_001073 | |
| Software, algorithm | LISA v1 | lisa.cistrome.org | | |

## Cell culture

### WI-38 maintenance and subculturing for replicative senescence time course-WT and hTERT

WI-38 cells were obtained from the Coriell Institute (AG06814-N) at PDL 15. The cells were grown in Dulbecco's modified Eagle's medium (DMEM, Gibco, 11885084) supplemented with 10% dialyzed fetal bovine serum (FBS, Sigma, F0392) and maintained in an incubator set to 37° C, 5% CO2, and 20% O2. 0.3–0.5 E6 WI-38 cells were seeded and maintained on 10 cm collagen coated plates (Corning, 354450) and split when cells reached about 70% confluence. For WI-38 cells PDL 15 tp PDL 40, it took about 4 days to reach 70% confluence. WI-38s > PDL 40 were slower growing, and therefore split every 5–7 days instead; media was replenished every 3–4 days (supplementary methods). Cells were passaged by washing the cell monolayer with PBS, followed by incubating cells with TrypLE Express (Gibco, 12604013) for 5 min. Media was added to neutralize the TrypLE and the cell suspension was collected into conical tubes. The cell suspension was spun at 200 x g for 5 min and the supernatant

was aspirated. The resulting cell pellet was resuspended in media and cells were counted with viability measurements on a ViCell XR Cell Analyzer (Beckman Coulter). Cell numbers taken from the ViCell were used to calculate population doublings (PDLs). The following formula was used to calculate PDLs:

$$PDL = \log2(\text{number of cells harvested}) \log2(\text{number of cells seeded})$$

All cells collected for assays were sampled in triplicate, at 2.5–3 days after seeding, targeting 60–70% confluence to avoid the confounding effects of confluence. Confluence levels were determined with a phase contrast EVOS microscope (ThermoFisher). Slower growing cells (PDL > 40) were sampled at 3.5–4 days after seeding. Cells were tested every month for mycoplasma contamination using the MycoAlert Mycoplasma Detection Kit (Lonza). A smaller subset of three PDL timepoints (PDL 45, PDL 55, PDL 56) were generated in a secondary time course to ensure 'deep' senescence (2–4 months after cell cycle cessation) was maintained as previously reported (*De Cecco et al., 2019*).

WI-38 hTERT cells were maintained using the same conditions as WT WI-38s, targeting 50–70% confluence for cell splitting and sampling alongside WT cells.

## Sampling proliferating WI-38 cells at different cell densities confounds assessment of cell cycle state and senescence

A limited pilot (performed prior to RS time course) experiment was conducted to determine optimal sampling conditions. Using gene expression as a readout for cellular state, we performed RNA-seq on PDL 20, PDL 29, and PDL 39 WI-38 cells. We plated an equal number of cells in triplicate for the three PDLs and cells were collected for RNA extraction after 4 days of growth. We looked for any significant changes in transcript levels between the three samples.

We observed two notable trends. First, we found stronger expression of genes associated with cell cycle and mitosis in the older PDL 39 cells compared to PDL 20 and PDL 29 cells. This was unexpected based on previous observations of slower growth and less active mitosis in cells closer to senescence (*Hayflick, 1965*). Second, the cells at PDL 20 and PDL 29 expressed genes associated with hypoxic stress (*Figure 1—figure supplement 5A*).

These data, in tandem with the observation that older cells grow at a slower rate, suggest that the highly proliferative younger cells (PDL 20 and 29) grow to higher cell density faster and therefore encounter earlier in real time, the various normal mitosis-inhibiting stresses, for example hypoxia, contact inhibition, and starvation that trigger cell cycle exit (*Dayan et al., 2009*).

To understand the effects of growth rates and cell density on gene expression, we performed a cell density study (CD) wherein young PDL cells were sampled at increasing levels of cell density to measure the effects of cell density and proliferation on gene expression (*Figure 1A*). We observed that based on the seeding density we employed (to minimize the stress of sparse plating) the sampling window we had originally targeted (4 days after seeding) was located in a volatile region with respect to proliferation rate and hypoxic stress using Cyclin B1 (*CCNB1*) and VegfA (*VEGFA*) expression as proxies for each, respectively (*Figure 1—figure supplement 5B*).

Accordingly, we modified the sampling window by choosing a time interval (2.5–3 days) prior to the onset of VEGF-A induction and CCNB1 repression (*Figure 1—figure supplement 5B*). For older, slower growing cells in the plateauing region of the replicative senescence process (PDL >40) we shifted the sampling window to 3.5–4 days based on visual inspection and cell counts. Ultimately in the actual extended time course, we tracked cell density with a phase contrast EVOS microscope to ensure cellular confluence was similar across all time points during sampling (*Figure 2—figure supplement 2*).

## WI-38 irradiation time course

WI-38 cells at PDL 20 were plated in DMEM supplemented with 10% FBS with 50,000 cells per well in a 6-well collagen coated plate. Cells were allowed to settle and adhere to plates. All cells were adhered to the TC plate by 2 hr and were subsequently treated with 10 Gy of X-rays (Faxitron CellRad). Cells were sampled between 1 day and 9 days for transcriptome profiling. Each time point was sampled in triplicate. RNA was extracted according to methods below.

## WI-38 cell density time course

0.3 E6 WI-38 cells at PDL 23 were seeded and grown on 10 cm collagen-coated plates in DMEM supplemented with 10% FBS. Cells were grown and sampled intermittently between 1 day and 10 days for transcriptome profiling. Samples at each time point were taken in triplicate. RNA was extracted according to methods below.

## SA-β GAL staining

Cells were stained for senescence associated beta-gal using the Senescence β-Galactosidase Staining Kit (Cell Signaling, 9860) by following the manufacturer's published protocols exactly.

## WI-38 hTERT cells and lentiviral transduction

293T cells were transfected with a lentiviral target plasmid expressing hTERT (pCDH-CMV-hTERT-EF1a-puro) and lentiviral packaging constructs overnight; 48 hr later, viral supernatant was collected. WI-38 cells were transduced with viral supernatant in the presence of 5 µg/mL polybrene and selected for 7 days with 1 µg/ml of puromycin.

## WI-38 + verteporfin treatment

A total of 200,000 WI-38 cells at PDL 38 were seeded onto each well of a six-well collagen-coated plate and grown with 2 ml of DMEM supplemented with 10% FBS. After 4 days of growth, cells were treated with 5 or 10 µM of verteporfin (RD Systems 5305/10) for 2 hr or DMSO. RNA was extracted after 2 hr of treatment according to methods described below.

## Bulk RNA-Seq methods

### RNA collection and library preparation

Total RNA was extracted from cells using the Direct-zol RNA Miniprep Plus kit (Zymo Research R2072) for all bulk RNA-seq time course experiments and verteporfin treatment experiment. Manufacturer's protocol was followed exactly and in-column DNAase digestion was performed. RNA quality score and concentration was measured using the Fragment Analyzer (Agilent 5200) with the Fragment Analyzer Standard Sense RNA kit (Agilent Technologies DNF-471–0500). All samples required to have a RIN score of >7 for processing. RNA sequencing libraries were prepared as directed using TruSeq Stranded mRNA Library Prep Kit (Ilumina 20020595), with 1000 ng of input material. Samples were amplified for 12 cycles of PCR with TruSeq RNA CD Index Plate (Illumina) and pooled. 3 nM libraries were loaded across four lanes on the HiSeq 4000 (Illumina).

### Read processing and quantification

Reads generated from the Illumina HiSeq 4000 were demultiplexed with bcl2fastq (version=2.20) based on the barcode sequence of each sample. Average read depth across samples was 50 million paired-end reads. Reads were pseudo-aligned and quantified using Salmon (version=0.8.2) by deploying the mapping based mode using a Salmon command 'index' with default parameters based on 10 X genomics hg38 transcriptome annotations optimized for single cell RNA-seq ('refdata-cellranger-GRCh38-3.0.0', cellRanger version 3) to ensure accurate comparison between bulk and single-cell RNA-seq (*Patro et al., 2017*). Annotations can be obtained running 'wget https://cf.10xgenomics.com/supp/cell-exp/refdata-cellranger-GRCh38-3.0.0.tar.gz'.

### Differential expression analysis

DESeq2 (version=1.30.1) was used for differential analysis of the RNA-seq data *Love et al., 2014*. Wald test was used to estimate fold change and significance using the model:~ time + batch, where time is a numeric variable representing the fraction of time course complete and batch is a categorical variable used only with replicative senescence with the addition of the 'deep' senescence time points. DEGs were defined as having FDR adjusted p-values < 0.01.

### Batch correction and hierarchical clustering

For clustering and visualization, we corrected the raw replicative senescence count table by batch before converting to transcripts per million (TPM) with a + 1 pseudocount and combining with radiation

induced senescence and cell density samples using Combat-seq (sva R package version=3.38.0) (*Zhang et al., 2020*). Significant genes from each condition were concatenated to generate a universe of significant change genes used for *Figure 1*. Each sample was then converted to log2 fold change vs. the mean of the initial time point.

## Gene set enrichment analysis

For gene set enrichment analysis, we downloaded the MSigDB Hallmark gene sets from misgDB website (version 7, gene symbols) *Liberzon et al., 2015*. For each time course (replicative senescence, radiation-induced senescence, and cell density) we we ranked all genes by the log2 fold change across time generated by DESeq2. Then GSEA was performed on the ranked gene set using the R package 'fgsea' version 1.16.0 which is an implementation of GSEA in R (*Subramanian et al., 2005*; *Korotkevich et al., 2019*; *Mootha et al., 2003*). By default, GSEA tests for enrichment of each gene set in each condition in both directions. We report the -log10 Benjamini-Hochberg corrected p-value in *Figure 1* and use the normalized enrichment score (NES) to assign direction of the change.

## Single-cell 3' RNASeq methods

### Cell collection

At each time point, singlet hTERT controls and experimental samples were processed with Chromium Single Cell 3' RNAseq kit V2 (10 x Genomics 120237) through cDNA amplification cleanup, where they were frozen at –20° C. Once all time points were converted to cDNA, the frozen cDNAs were thawed and batched for library construction. The following modifications were made to the process: Reverse transcription reactions were brought up to volume with DMEM +10% FBS instead of water, each emulsion targeted 3000 cells (5200 cells loaded), and cDNA was amplified 12 cycles, with 12 cycles of index PCR. There was no hTERT control for time point 3, and one of the replicates for PDL 25 dropped out during library construction. Remaining samples were pooled at equimolar concentration and sequenced on a HiSeq4000 with the standard 26,8,0,98 run base pairs per read configuration.

### Single-cell data processing, normalization, scoring, clustering, and DEG analysis

The raw single-cell reads were demultiplexed by sample using bcl2fastq. Alignment, cell barcode demultiplexing, transcript quantification and sample merging were carried out with CellRanger 3.0 using the hg38 CellRanger 3.0 gene annotation ('refdata-cellranger-GRCh38-3.0.0', cellRanger) with default options and parameters. Filtered cell by gene matrices were normalized using Seurat (version = 4.0.1.9005) and SCTransform (version = 0.3.2) (*Hafemeister and Satija, 2019*; *Stuart et al., 2019*). Dimension reductionality was carried out with PCA (n = 50). Clusters were defined using the louvain algorithm resolution = and cells were visualized with UMAP projection *McInnes et al., 2018*; *Becht et al., 2018*; *Blondel et al., 2008*. For cell cycle scoring and phase determination as well as senescence scoring, we applied the Seurat implementation of the scoring function as previously described (*Nestorowa et al., 2016*).

DESeq2 was used for differential gene expression analysis to identify significantly changing genes within individual clusters as a function of increasing PDL (PDL by mitotic phase) (*Figure 3D*) using a an input matrix of gene counts by cells per PDL by mitotic phase. Gene counts were summarized across cells for each mitotic phase (PDL 50 removed). To expedite computation, we restricted analysis to highest expressed 8000 genes. For each PDL by cell cycle phase grouping, we required that >15 representative cells must exist to be considered in analysis. To visualize differentially expressed genes, we converted single cell counts to counts per million ( + 1 pseudocount) and averaged across PDL by cell cycle phase and calculated the log2 fold change at each PDL by cell cycle phase against the earliest PDL for that cell cycle phase using a concatenated list of all significantly changing genes generated by DESeq2 from each cell cycle phase.

### Pseudotime analysis

For pseudotime analysis of the single-cell data, we used the R package monocle3 (version=1.0.0) which implements PCA, Leiden clustering, and UMAP prior to partitioning and trajectory analysis. We first isolated WT cells by removing hTERT cells and those not belonging to the primary grouping. To focus trajectory analysis on the replicative senescence progression, we used only the top genes

found to significantly change with replicative senescence in bulk RNA-seq for PCA (padj <0.01) and log2 fold change >3 or log2 fold change < −3). In applying the Monocle trajectory analysis: (learn_graph(cds,close_loop = FALSE,learn_graph_control = list(minimal_branch_len = 35, geodesic_distance_ratio = 0.5, euclidean_distance_ratio=1)), we designated cycling cells at the opposite end of UMAP project from the senescent cells as the start point manually (*Trapnell et al., 2014*; *Qiu et al., 2017*; *Cao et al., 2019*; *Traag et al., 2019*; *Levine et al., 2015*; *McInnes et al., 2018*). After establishing a trajectory, we then employed the Monocle3 'graph test' function to isolate genes that significantly change as a function of pseudotime. Pseudotime estimation was output from monocle3 using the learn graph function for building a trajectory. Smoothed pseudotime trajectories use for *Figure 7C* were calculated for significantly changing genes (with pseudotime) by binning cells across pseudotime into 60 bins and using a cubic spline to estimate expression at each bin. For each gene, the smoothed trajectory was set from 0 (minimal) to (1) maximal expression. Genes were organized with K-median clustering (k=25) using cosine similarity. For visualization the median expression value for each pseudotime bin and cluster was calculated. Genes from each cluster were fed into LISA TF analysis (below).

## ATAC-SEQ Methods

### ATAC-Seq library preparation and sequencing

Freshly harvested cells were used for all reactions. Briefly, the cell monolayer was washed with PBS, trypsinized with TrypLE Express (Gibco 12604013), resuspended in media, and cells were pelleted. Cells were counted and 100,000 cells were used in each reaction. Cell lysis, DNA transposition, and library construction was followed from the Omni-ATAC protocol (*Corces et al., 2017*). Libraries were amplified for 13 total cycles. Sample purification and size selection were performed on the Pippin high throughput size selection platform (Sage Science) using 2% agarose cassettes to isolate nucleosome free fragments that were <300 base pairs in library fragments and thus corresponding to <150 base pairs fragments after accounting for sequencing adaptors. Quality of ATAC-seq libraries were assessed with the Agilent Bioanalyzer 2,100 with DNA high sensitivity chips (Agilent Technologies 5067–4626). Libraries were sequenced on the HiSeq4000 with paired end sequencing using 2 × 150 bp reads (Illumina).

### ATAC-Seq data processing, peak calling, and differential accessibility

We first trimmed the raw fastq files with cutadapt (version=2.4) to remove standard Nextera PE adapters:

> adapt -a file:$ADAPTER -A file:$ADAPTER -o $SAMPLE.R1.fq -p $SAMPLE.R2.fq –pair-filter=any –minimum-length=30 $R1 $R2
> Then we aligned with bowtie2 (version version 2.3.4.1) to align the trimmed reads to hg38:
> bowtie2 -x $INDEX –1 $SAMPLE.R1.fq –2 $SAMPLE.R2.fq –no-mixed –no-discordant -X 1000

After alignment, we used samtools (version=1.2) flags (-f 0 x02 and -q2 0) to filter for only properly paired and high quality reads. PCR duplicates are removed using picard (version=2.6.4) MarkDuplicates. Finally for each bam file, we adjusted the reads ends by Tn5 offset ( + 4 on+ strand, −5 on -strand).

ATAC-seq QC (mitochondrial percent and transcription start site score) and alignment metrics were generated with R package ATACseqQC (version=1.14.4) and multiQC respectively (version=1.11) (*Ewels et al., 2016*) using TxDb.Hsapiens.UCSC.hg38.knownGene (version = 3.10.0) annotations for transcription start site score calculation.

For peak calling, we created a condition specific peak atlas by pooling all replicates in a specific condition and applied macs2 (version = 2 2.1.2) for peak calling on the pooled bam file with options (-g hs -p 1e-1 –nomodel –shift –37 –extsize 73) (*Zhang et al., 2008*). In addition, we performed peak calling on each individual replicates as well. Then we performed irreproducible discovery rate analysis on each condition specific peak atlas for each pair of replicates and filter for peaks that are reproducible in at least two replicates (IDR threshold of 0.05). A single accessibility atlas is created by merging condition-specific peak atlas across all conditions (*Figure 5—source data 3*). Peaks were assigned to nearest gene if it is within 50 kb, otherwise it is annotated as intergenic.

Read count was performed using countOverlaps function from R package GenomicRanges (version = 1.42.0) (*Lawrence et al., 2013*). We performed quantile normalization of the count matrix using normalize.quantiles function of R package preprocessCore (version=1.52.1).

Limma-voom (Limma version = 3.46.0) was used for differential accessibility analysis (*Ritchie et al., 2015*). Fold change and fdr adjusted p-values were estimated using moderated t-test statistic based on the model: time+ condition + time:condition. We performed separate tests for time point one versus each of the other time points.

## ATAC-Seq chromatin state and NAD/LAD analysis

To quantify ATAC-seq reads in chromatin states, we retrieved hg38 ENCODE IMR-90 chromatin state labels from (*Ernst and Kellis, 2015*; *Figure 5—source data 1*). Next, we quantified coverage for each instance of all 25 chromatin states genome wide using countOverlaps function from R package GenomicRanges (*Lawrence et al., 2013*). We performed quantile normalization of the chromatin state count matrix using normalize.quantiles function of R package preprocessCore (*Figure 5A*).

ATAC-seq peaks were assigned to chromatin states using the findOverlapsOfPeaks function from the R package GenomicRanges (*Lawrence et al., 2013*). We used only significantly ($P < 0.001$) changing peaks(with senescence) from LIMMA-voom analysis. To simplify the overlap of these two sets of genomic intervals, we took only peaks that fell within ('inside') or encompassed a chromatin state annotation ('inside feature'). Peaks encompassing more than one chromatin state interval were discarded (*Figure 5A and B*).

For analysis of NAD and LAD domain overlap we collected IMR-90 NAD labels from *Dillinger et al., 2017* and IMR-90 LAD labels from *Sadaie et al., 2013* (*Figure 5—source data 5*). For calculating overlap Z-score between genomic intervals sets (e.g. NADs/LADs vs. gene annotation) (TxDb. Hsapiens.UCSC.hg38.knownGene, version = 3.10.0) for Figure S18C we used the overlapPermTest function from the R package regioneR (version = 1.22.0) *Gel et al., 2016*.

For testing overlaps between NAD/LAD domains and significantly changing genes, we used the overlapPermTest function from the regioneR package using the top induced genes from each bulk RNA-seq condition (RS,RIS, and CD) against a universe of all genes sampled from the same expression distribution to control NAD and LAD bias across different levels of expression. For gene annotations, we used TxDb.Hsapiens.UCSC.hg38.knownGene (version = 3.10.0).

## Metabolomics

### Extraction of water-soluble metabolites from lung fibroblast cell culture

Twenty-four hours before metabolite extraction, the medium was aspirated, cells were washed with unconditioned medium and then the medium was replaced. For metabolite extraction, cells were washed once in 37°C warm PBS-buffer immediately followed by the addition of 3.5 mL of freezer-cooled (-20) LC-MS grade 80:20 MeOH/H2O (Sigma Aldrich). The plates were then held at -20 for 2 hr, then harvested with a sterile cell scraper while at –20°C and transferred to -20 cold 5 mL centrifuge tubes (Eppendorf Lo-Bind). After centrifuging the cell-extracts in a 4 centrifuge for 5 min at 2000 x g, the supernatants were transferred into new cold centrifuge tubes and dried under nitrogen at 4. Dried extracts were stored at -20.

### LC-MS/MS analysis of cell culture extracts

Dried supernatants were resuspended in 200 µL of water containing 1 µg/mL of deuterated lysine and deuterated phenylalanine and 250 ng/mL of deuterated succinate (Sigma Aldrich) as internal standards. For negative ion mode, the resuspended samples were diluted 1:4 in water, for positive ion mode, they were diluted 1:4 in acetonitrile. Samples were then centrifuged at 18,000 x g for 5 min, the supernatant was moved to HPLC vials and 5 µL was injected for analysis by LC-MS on Vanquish UPLCs (Thermo Scientific) coupled to Q Exactive Plus mass spectrometers (Thermo Scientific).

For analysis in negative ion mode, separation of compounds was achieved by reverse-phase liquid chromatography on a Zorbax Extend C18 column (150 × 2.1 mm, 1.8 µm particle size [Agilent 759700–902]). Mobile phase A was 10 mM Tributylamine and 15 mM Acetic Acid in 97:3 water:methanol at pH 4.95 and mobile phase B was methanol. Prior to injection, the column was equilibrated in 0% B for 4 minutes. The gradient eluted isocratically in 0% B for 2.5 min, increased to 20% B over 2.5 min, held at 20% B for 2.5 min, increased to 55% B over 5.5 min, increased to 95% B over 2.5 min, maintained at

95% B for 3 min, then decreased to 0% B over 0.5 min, where it was held for 3 min, resulting in a total run time of 26 min. The LC separation was carried out at 30 column temperature using a 200 µL/min flow rate. For MS-analysis, parameters on MS1 were set to 70,000 resolution with an AGC target of 1e6 at a maximum IT of 100ms. The scan range was 70–1050 m/z. MS2 parameters were set to 17,500 resolution at loop count 6, an AGC target of 1e5 at a maximum IT of 50ms, an isolation window of 1 m/z and an underfill ratio of 1%. Dynamic exclusion was set at 20 s with an apex trigger from 3 to 12 s. Stepped collision energies were set to 20, 50, and 100% NCE.

For analysis in positive ion mode, compounds were separated via hydrophilic liquid interaction chromatography (HILIC), using a SeQuant ZIC-pHILIC column (150 × 2.1 mm, 5 µm particle size) Millipore (150460). Mobile phase A consisted of 20 mM ammonium bicarbonate at pH 9.2 in $H_2O$, and mobile phase B was acetonitrile. Prior to injection, the column was equilibrated for 6 minutes in 80% B. The gradient then decreased to 20% B over 20 min, then to 15% B over 2 min, returned to 80% B over 0.5 min and held there for 1.5 min for a total run time of 30 min. The column temperature was 35 with a flow rate of 150 µL/min. For MS-analysis, the MS1 parameters were as described for negative ion mode except the AGC target was 3e6. MS2 parameters were the same with following exceptions: dynamic exclusion was set to 25 s with an apex trigger from 3 to 10 s. Stepped collision energies were set to 20, 40% and 80% NCE.

## Metabolomics batch correction and differential analysis

Protein lysates derived from same plates used for metabolomics collection were assayed for total protein concentration with Pierce BCA Protein Assay Kit (ThermoFisher 23225) for normalizing raw metabolite values. The median protein concentration for each PDL and PDL.ctrl was calculated and divided by the median protein concentration across all samples for either cell line (WT and hTERT) to derive a protein concentration normalization factor. The raw metabolite values for each PDL and hTERT PDL.ctrl were multiplied by the protein concentration normalization factor for that strain and sample. Protein normalized metabolite values were then converted to log2 fold change values using the first time point as a reference (*Figure 3—source data 2*). To remove sample day batch effects, we took the protein normalized metabolite values and divide each WT metabolite value by the corresponding metabolite value in the temporally paired hTERT sample (e.g. PDL 25 values were divided by PDL 25.ctrl values) and converted to log2 fold change with first sample as reference.

Finally, the normalized log2 ratios were fit to a linear model to test for a linear trend across the time course. For each metabolite, we fit the model $y_{it} = \beta_0 + \beta_1 T_t + \epsilon_{it}$, where i = (1,...,3) indexes the replicates for each time point and $t$=(1,..,4) indexes the cell passage vector T = (33, 37, 46, 50).

The model was fit with the lm() function in R Version 4.1.2. p-values from the two-sided hypothesis test $H_0 : \beta_1 = 0$ were FDR adjusted (across all proteins) using the qvalue function from the qvalue R package (version =2.26.0). If present, asterisks in figures indicate that the FDR adjusted p-value from this test was <.05.

## Proteomics methods

### Materials and sample preparation, extraction and digestion

LC-MS grade organic solvents, water, and tandem mass tag (TMT) isobaric reagents were purchased from Thermo Fisher Scientific (Waltham, MA). Trypsin was ordered from Promega Corporation (Madison, WI) and Lys-C from Wako Chemicals USA (Richmond, VA). Sep-Pak C18 cartridges were from Waters Corporation (Milford, MA). Unless otherwise stated, all other chemicals were purchased from Sigma-Aldrich (St. Louis, MO).

At time of sample collection, cells were trypsinized off the monolayer with TrypLE Express and media was used to neutralize the reaction. Cells were pelleted at 200 x g for 5 min and the supernatant was aspirated. Cell pellets were flash frozen in liquid nitrogen and stored at –80°C until all samples were collected and ready for proteomics analysis. Cell pellets were resuspended in 450 µL of lysis buffer (75 mM NaCl, 3% SDS, 50 mM HEPES, pH 8.5) and lysed by passage through a BD PrecisionGlide 21-gauge needle (20 X). The lysate was sonicated for 5 min and then centrifuged (5 min, 16,000 x g) to remove cellular debris and the supernatant was collected.

Proteins were reduced with 5 mM dithiothreitol (DTT) for 30 min at 56 with shaking. Proteins were then alkylated with 15 mM iodoacetamide (IAM) for 30 min at room temperature (RT) in the dark, and excess IAM was quenched with 5 mM DTT for 20 min at RT in the dark. Protein purification was

accomplished using a methanol-chloroform precipitation. Briefly, 800 µL methanol, 200 µL chloroform and 600 µL water were sequentially added to 200 µL of cell lysate, with 5-s intervals of vortexing between each addition. The samples were centrifuged for 30 min (16,000 x g at 4°C) to induce phase separation and both the top and bottom layers were removed. The precipitated protein pellet was washed with 600 µL methanol, vortexed briefly, then centrifuged for 10 min (16,000 x g at 4 °C). The methanol layer was removed and protein pellets were dried at RT for 10 minutes. Protein pellets were resuspended in digestion buffer (8 M urea, 50 mM HEPES, pH 8.5). The urea concentration was diluted to 4 M, then proteins were digested with Lys-C overnight (10 ng/µL, 25, 16 hr). The urea concentration was further diluted to 1 M and samples were digested with trypsin (5 ng/µL) for 6 hr at 37 ∘C.

Following digestion, peptides were acidified with trifluoroacetic acid (TFA) to a final concentration of 0.5% TFA. Peptides were desalted using Sep-Pak C18 solid-phase extraction (SPE) columns and samples were eluted sequentially, first with 40% acetonitrile (ACN)/0.5% acetic acid and then 80% ACN/0.5% acetic acid. Eluted peptides were dried in a CentriVap Benchtop Vacuum Concentrator (Labconco, Kansas City, MO) running at 30 °C. Peptide concentrations were measured using the Pierce BCA Protein Assay Kit, then 50 µg aliquots of each samples were dried in the CentriVap for further processing.

## Tandem mass tag (TMT) labeling

Dried peptides were resuspended in 50 µL 200 mM HEPES/30% anhydrous ACN, then 200 µg of each TMT tag was added to 50 µg peptides. TMT 131 c was used as the 'bridge sample' while the other tags (126, 127 n, 127 c, 128 n, 128 c, 129 n, 129 c, 130 n, 130 c, 131 n) were used to label the individual samples. The TMT reaction was incubated for 1 hour at room temperature with gentle shaking, then quenched with 11 µL 5% hydroxylamine/200 mM HEPES. All samples were acidified to a final concentration of 0.5% TFA. A small amount (4 µL) of each labeled sample was combined and desalted using StageTips to check TMT ratios and labeling efficiency. The TMT-labeled samples were then combined at a 1:1:1:1:1:1:1:1:1:1:1 peptide ratio into 11-plex samples (*Rappsilber et al., 2007*). The combined samples were desalted using Sep-Pak C18 cartridges and dried under vacuum.

## High pH reversed-phase (HPRP) fractionation

The pooled TMT-labeled peptides were fractionated using high pH reversed-phase liquid chromatography on an Agilent 1260 Infinity HPLC equipped with a diode array detector set at 215, 220, and 254 nm wavelengths (Agilent Technologies, Santa Clara, CA). Peptides were separated on an Agilent ZORBAX Extend-C18 column (4.6 mm x 250 mm, 5 µm particle size) running at 500 µl/min at 25°C. Peptides were eluted with a gradient with initial starting condition of 100% buffer A (5% ACN, 10 mM ammonium bicarbonate) and 0% buffer B (95% ACN, 10 mM ammonium bicarbonate). Buffer B was increased to 35% over 60 min, then ramped up to 100% B in 6 s where it was held for 5 min. Buffer B was then decreased to 0% over 6 s and held for 10 min to re-equilibrate the column to original conditions. The samples were fractionated into 96 fractions, then pooled into 12 fractions as previously described (*Huttlin et al., 2010*). The fractions were dried under vacuum and resuspended in 5% ACN/5% formic acid (FA) for LC-MS/MS analysis.

## Proteomics data acquisition and analysis

LC-MS/MS Data Acquisition All samples were analyzed by an Orbitrap Fusion Lumos Tribrid mass spectrometer coupled to an EASY-nLC 1200 (Thermo Fisher Scientific). Peptides were separated using a microcapillary column (100 µm x 250 mm long, filled in-house with Maccel C18 AQ resin, 1.8 µm, 120 Ã; Sepax Technologies, Newark, DE) operating at 60 °C with a flow rate of 300 nL/min. Peptides were eluted into the mass spectrometer using a 180 min method, with acetonitrile increasing from 6% to 30% over a 165 min linear gradient in 0.125% formic acid. Mass spectrometry data was collected in data-dependent acquisition (DDA) mode. A high-resolution MS1 scan (500–1200 m/z range, 60,000 resolution, AGC 5 × 105, 100ms max. injection time, RF for S-lens 30) was collected in the Orbitrap, and the top 10 precursors were selected for MS2 and MS3 analysis. Ions were isolated using a 0.5 m/z window for MS2 spectra. The MS2 scan was performed in the quadrupole ion trap (CID, AGC 1 × 104, 30% normalized collision energy, 35ms max. injection time) and the MS3 scan was analyzed in the Orbitrap (HCD, 60,000 resolution, max. AGC 5 × 104, 250ms max. injection time, 50% normalized collision energy). The max. cycle time was set at 5 s. For TMT reporter ion quantification, up to six

fragment ions from each MS2 spectra were selected for MS3 analysis using synchronous precursor selection (SPS).

## Proteomics data analysis

The *Ting et al., 2011* software pipeline and methods developed in the the Haas and Gygi labs was used to process all proteomics data (*Ting et al., 2011*). Raw files were converted to mzXML files and searched against a composite human UniProt database containing forward and reverse sequences using the Sequest algorithm. MS/MS spectra were matched with fully tryptic peptides from this composite dataset using a precursor ion tolerance of 20 ppm and a product ion tolerance of 0.6 Da. TMT modification of peptide N-termini and lysine residues ( + 229.162932 Da) and carbamidomethyl-ation of cysteine residues ( + 57.02146 Da) were set as static modifications. Oxidation of methionine residues ( + 15.99492 Da) was set as a variable modification. Peptide spectral matches were filtered to a 1% false discovery rate (FDR) using linear discriminant analysis (LDA) as previously described (*Ting et al., 2011*). Non-unique peptides that matched to multiple proteins were assigned to proteins that contained the largest number of matched redundant peptides sequences using the principle of Occam's razor (*Ting et al., 2011*).

Quantification of TMT reporter ion intensities was performed by extracting the most intense ion within a 0.003 m/z window at the predicted m/z value for each reporter ion. TMT spectra were used for quantification when the sum of the signal-to-noise for all the reporter ions was greater than 200 and the isolation specificity was greater than 0.75.

Peptide level intensities were converted to log2 ratios by dividing each scan by the intensity in the bridge channel. Relative protein abundance, for each sample, was estimated with the posterior mean from a previously described Bayesian model using methods as described by *O'Brien et al., 2018* with code located at https://github.com/ColtoCaro/compositionalMS (*O'Brien, 2020*).

Finally, the relative abundance estimates were fit to a linear model to test for a linear trend across the passages as described by *Gaun et al., 2021*. For each protein, we fit the model $y_{it} = \beta_0 + \beta_1 T_t + \epsilon_{it}$, where $=(1,...,3)$indexes the three replicates for each time point and $t=(1,...,7)$indexes the cell passage vector $T=(18, 25, 32, 33, 37, 46, 50)$.

The model was fit with the lm() function in R Version 3.6.0. p-Values from the two-sided hypothesis test $H_0 : \beta_1 = 0$ were FDR adjusted (across all proteins) using the p.adjust() function. If present, asterisks in all figures indicate that the FDR adjusted p-value from this test was <0.01.

## Regulatory analysis

### LISA TF analysis

Identification of putative regulators of the gene expression changes observed in the bulk RNA-seq experiments and the pseudotime analysis was carried out using (*Qin et al., 2020*); a recent algorithm built to leverage the vast amount of protein-DNA interactions catalogued via ENCODE. We used the GUI hosted at http://lisa.cistrome.org/ to enter lists of genes derived from K-medians pseudo-time clustering clustering. The online GUI has a max input of 500 genes. Accordingly, for clusters containing >500 genes, each gene was ranked by correlation with the cluster median and the top 500 genes were used. LISA output consists of a ranked file of transcription factors and chromatin modifiers with enrichment p-values associated with specific ENCODE experiments. Given that the calculated p-values were derived from gene sets of different sizes and from different number of supporting experiments, we used the ranks as the input parameter for the LISA analysis shown in *Figures 6D and 7D*. A universe of top factors across clusters was compiled by concatenating the top 5 TFs from each individual LISA output. The rank matrix was then centered across clusters to identify TFs with the highest rank in specific clusters.

### ATAC-Seq peak motif enrichment

We built a binary peak-by-motif matrix where each row is a binary vector representing the presence of 405 motifs we used from CisBP (Catalog of Inferred Sequence Binding Preferences) which is a freely available online database of transcription factor (TF) binding specificities: http://cisbp.ccbr.utoronto.ca (*Weirauch et al., 2014*). We only used motifs for TFs with measurable RNA-seq in WI-38 cells. In order to characterize transcription factor activity changes during senescence, we implemented both a gene-centric and a gene-indepedent approach.

In the gene-centric approach, we tested enrichment of each of the 405 motifs in peaks that are (1) associated with a gene set of interest (within 50 kb), and (2) significantly associated with senescence (adjusted p-value < 0.001 in PDL 50 vs. PDL 20) as compared to all the other peaks using binomial test.

Alternatively, we trained a ridge logistic regression model using the binary motif matrix as features to distinguish peaks of significantly increased accessibility during senescence from peaks of significantly reduced accessibility in PDL 50 vs. PDL 20. A model trained on two thirds of the data distinguishes the two sets of peaks with AUC=0.67 on held-out peaks. We then trained 10 independent models using all the data to evaluate the coefficients to identify features (motifs) that are most predictive of senescence.

## Acknowledgements

The authors thank Ari Firestone, Jason Rogers, Nick Bernstein, Jeff Settleman, Rochelle Buffenstein, Emily Stoops, Magdalena Preciado Lopez, and Antoine Roux for their feedback on the experiments, analysis and the manuscript. In addition the authors also thank Leonard Hayflick for guidance and discussion.

## Additional information

### Competing interests

Michelle Chan, Han Yuan, Ilya Soifer, Tobias M Maile, Rebecca Y Wang, Andrea Ireland, Jonathon J O'Brien, Jérôme Goudeau, Leanne JG Chan, Twaritha Vijay, Adam Freund, Cynthia Kenyon, Bryson D Bennett, Fiona E McAllister, David R Kelley, Margaret Roy, Robert L Cohen, Arthur D Levinson, David Botstein, David G Hendrickson: is affiliated with Calico Life Sciences, LLC. The author has no other competing interests to declare.

### Funding

No external funding was received for this work.

### Author contributions

Michelle Chan, Conceptualization, Data curation, Formal analysis, Investigation, Methodology, Project administration, Visualization, Writing - original draft, Writing - review and editing; Han Yuan, Data curation, Formal analysis, Methodology, Visualization; Ilya Soifer, Data curation, Formal analysis, Investigation, Methodology, Software, Visualization; Tobias M Maile, Jonathon J O'Brien, Formal analysis, Investigation, Methodology; Rebecca Y Wang, Andrea Ireland, Jérôme Goudeau, Leanne JG Chan, Twaritha Vijay, Investigation, Methodology; Adam Freund, Cynthia Kenyon, Bryson D Bennett, Fiona E McAllister, David R Kelley, Project administration, Supervision; Margaret Roy, Conceptualization, Investigation, Methodology, Project administration, Supervision; Robert L Cohen, Conceptualization, Project administration, Supervision; Arthur D Levinson, Conceptualization, Project administration, Resources, Software; David Botstein, Conceptualization, Project administration, Resources, Supervision, Writing - review and editing; David G Hendrickson, Conceptualization, Data curation, Formal analysis, Investigation, Methodology, Project administration, Supervision, Visualization, Writing - original draft, Writing - review and editing

### Author ORCIDs

Jonathon J O'Brien ![ORCID] http://orcid.org/0000-0001-9660-4797
Jérôme Goudeau ![ORCID] http://orcid.org/0000-0002-2483-1955
Adam Freund ![ORCID] http://orcid.org/0000-0002-7956-5332
Cynthia Kenyon ![ORCID] http://orcid.org/0000-0003-3446-2636
David Botstein ![ORCID] http://orcid.org/0000-0001-9499-4883
David G Hendrickson ![ORCID] http://orcid.org/0000-0002-1884-5234

### Decision letter and Author response

Decision letter https://doi.org/10.7554/eLife.70283.sa1
Author response https://doi.org/10.7554/eLife.70283.sa2

## Additional files

### Supplementary files
• Transparent reporting form

### Data availability
Sequencing data have been deposited in GEO under accession code GSE175533. Proteomics data have been deposited to the ProteomeXchange Consortium via the PRIDE partner repository under accession code EBI PRIDE Code for processing and analyzing data modalities have been deposited at https://github.com/dghendrickson/hayflick, (copy archived at swh:1:rev:e32e36ceb9aed-82f15b794b28bbbcee51bf5e9e8). Source data for figures and analysis have been deposited at https://github.com/dghendrickson/hayflick and/or uploaded as source data files.

The following datasets were generated:

| Author(s) | Year | Dataset title | Dataset URL | Database and Identifier |
|---|---|---|---|---|
| Chan M, Hendrickson DG | 2021 | Revisiting the Hayflick Limit: Insights from an Integrated Analysis of Changing Transcripts, Proteins, Metabolites, and Chromatin | https://www.ncbi.nlm.nih.gov/geo/query/acc.cgi?acc=GSE175533 | NCBI Gene Expression Omnibus, GSE175533 |
| Chan M, Hendrickson DG | 2021 | Revisiting the Hayflick Limit: Insights from an Integrated Analysis of Changing Transcripts, Proteins, Metabolites, and Chromatin | https://www.ebi.ac.uk/pride/archive/projects/PXD028799 | PRIDE, EBI PRIDE |

The following previously published datasets were used:

| Author(s) | Year | Dataset title | Dataset URL | Database and Identifier |
|---|---|---|---|---|
| Gregory BD, Gosai S, Crowe EP, Torres CA | 2014 | Transcriptome analysis of oxdative-stress induced senescence in human astrocytes | https://www.ncbi.nlm.nih.gov/geo/query/acc.cgi?acc=GSE58910 | NCBI Gene Expression Omnibus, GSE58910 |
| Németh A, Dillinger S | 2017 | Mapping of nucleolus-associated chromosomal domains during replicative senescence | https://www.ncbi.nlm.nih.gov/geo/query/acc.cgi?acc=GSE78043 | NCBI Gene Expression Omnibus, GSE78043 |
| Salama RA, Narita M | 2013 | Redistribution of the Lamin B1 genomic binding profile affects spatial rearrangement of heterochromatic domains and gene expression during senescence | https://www.ncbi.nlm.nih.gov/geo/query/acc.cgi?acc=GSE49341 | NCBI Gene Expression Omnibus, GSE49341 |

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
