## [Editor Report]

This manuscript reports a unique, comprehensive, multi-omic resource for the study of replicative senescence. This resource encompasses temporal metabolomic, proteomic, bulk transcriptomic, single cell transcriptomic, and chromatin accessibility states of fibroblasts as they transition from proliferative to replicatively senescent. Hence, it will be considered a valuable resource by aging researchers.

---

## [Decision Letter]

**Decision letter after peer review:**

Thank you for submitting your article "Revisiting the Hayflick Limit: Insights from an Integrated Analysis of Changing Transcripts, Proteins, Metabolites and Chromatin" for consideration by *eLife*. Your article has been reviewed by 3 peer reviewers, and the evaluation has been overseen by a Reviewing Editor and Matt Kaeberlein as the Senior Editor. The following individuals involved in review of your submission have agreed to reveal their identity: Lynne Cox (Reviewer #2); Payel Sen (Reviewer #3).

Essential revisions:

1) All three reviewers have raised various concerns about robustness of analyses and statistics for the multi-omics data. They recognize that significant amounts of these analyses will need to be redone and some additional analyses were recommended. However, they all agree that these revisions are essential. Please address these concerns as much as possible.

2) Reviewers #2 and #3 pointed out the necessity to provide some biological validations for novel findings from these multi-omics analyses. This will require additional experiments but will greatly improve this manuscript.

3) All three reviewers have identified the lack of description for certain methods and techniques, mislabeling, typographical errors, etc. Please address all these issues.

4) Reviewer #2 pointed out that the proteomics data should be made publicly available at one of the major proteomics data repositories. Reviewer #1 also pointed out that in-house software codes and scripts for data analyses should also be made publicly available.

*Reviewer #1 (Recommendations for the authors):*

1. There are major concerns about the statistics in the analysis of the dataset.

a. For the differential expression analysis (4.2.3), the authors state: "DEGs were defined as having p-values < 0.001." This is an incorrect method of filtering DEGs as it does not correct for multiple hypothesis testing. This is all the more surprising because DESeq2, used by the authors, natively provides FDR corrected p-values. The data should be re-analyzed after proper multiple hypothesis correction control (native DEseq2 FDR, Bonferroni, or other).

b. Potentially related to the absence of FDR thresholding, the number of DEGs identified-"8969 genes change with increasing PDL … 11652 genes vary with increasing cell density"-seemed, to this reviewer, really high, and likely the result of improper FDR correction. It will be interesting to see how those numbers change with the multiple testing correction.

c. Based on the methods, the other analyses should also always incorporate a multiple testing correction, i.e. differential ATACseq peaks (4.4.3 p < 0.001, no FDR correction), metabolomics (4.5 no information on differential analysis) and proteomics (4.6.5 no information on differential analysis).

d. When carrying out differential analyses, the authors should always avoid filtering on fold change. They do this for instance in section 2.4 "we focused on compounds exhibiting PDL-dependent changes in abundance > log2 0.5.". Fold-change filtering leads to extremely poor FDR control and should not be used post hoc – the only acceptable method to integrate fold change filtering includes a new statistical framework which takes into account fold change for FDR calculations (see PMC2654802). All results should be presented with this criterion removed or re-calculated with a method that takes fold-change into account.

e. Based on figure legends (Figure 1F, 3A, etc) and methods (4.2.5), FDR correction for multiple testing was also not performed for GSEA results. Please recompute everything including the correction, and only report corrected p-values.

f. There are several key issues with the ATAC-seq analysis (4.4.2), which will require a ground-up reanalysis of the data. Quantile normalization of count data and subsequent use of limma for differential accessibility analysis of ATAC-seq (4.4.2, end) is potential problematic. Indeed, limma is not appropriate for count data (it should only be used for continuous data such as microarray) unless the authors mean that they used limma-voom, which is a different algorithm. Please correct this in the revised analyses. In cases where differential depth cannot be corrected for appropriately, the authors should instead use downsampling of mapped reads to match the lowest depth sample. This will remove sequencing depth biases and allow proper count based methods, such as DEseq2, to perform properly.

2. Additional methodological information is needed for long-term reproducibility of analyses.

a. For reproducibility of code and analyses, all analytical code for this study should be deposited in a repository such as github or made available as a Supplementary file.

b. Please include all version numbers for all used software (e.g. R, Trimmomatic, Salmon, LISA, etc.) packages and R packages (e.g. DESeq2, fgsea, etc.) command parameters where relevant (such as for Trimmomatic, if not available elsewhere). The same should be applied to annotation databases, and if a version number doesn't exist, date of access should be provided.

c. All reagents (e.g. DMEM) should include suppliers and catalogue numbers in order to remove any ambiguity as to which product was used.

d. Salmon is a pseudoaligning method that uses a transcriptome reference, not a genome. The methods say that the index was derived from hg38, but how and using which information and software is not clear. In general, the authors should make sure that all key analytical steps are explicitly documented in the methods for long term reproducibility of the study.

3. It is completely unclear which data/plots were generated using the Illumina mRNA kit and which were generated using the Total RNA Kit. Please specify in text and legend.

*Reviewer #2 (Recommendations for the authors):*

The -omics aspects of the paper are generally well conducted and provide either confirmatory data on pre-existing studies of senescence or add new data (especially on the key FMT pathway and transcription factors involved). Log2FC of +/- 0.5 is somewhat lower than usual cut-offs, so justification is needed where that is selected. No mention is made of quality control steps in RNAseq analysis, but I assume these were conducted? Use of in-house programs for proteomics analysis does mean that others will not be able to conduct the same analyses, though availability of datasets should allow for validation of findings on other data analysis platforms – have these been publicly deposited as links provided are only to RNAseq datasets? From the protocol provided, it is also not clear that the methodology for protein extraction would release chromatin-bound factors – inclusion of a potent nuclease e.g. Benzonase, would ensure that tightly bound proteins are not lost in the spin steps to remove 'cellular debris'.

Figure 2E: there appears to be an aberrant pattern for cells at PDL33 that is present in all parts – is 33 an outlier? What is this cluster?

The cell biology aspects are much less well described making it impossible for others to reproduce the work – in particular, cell seeding density is critical but not even mentioned for longitudinal culture, and Figure 1D images suggest significant cell crowding. Along the senescence trajectory, primary fibroblasts grown in monolayers increase in surface area (and to some extent in volume) and populations at later PDLs undergo contact inhibition at lower cell densities than those at earlier PDLs so it may be the case that some of the 'gradual' changes (especially those associated with cell cycle changes) reflect simply quiescence form contact inhibition rather than steps to senescence. Methodology to count cells and calculate PDL is needed as different labs may conduct this differently. Daily microscopy inspection of cultures to assess appropriate time points for subculturing is essential – simply choosing to split at 4 or 7 days according to PDL does not account for the morphological changes that occur across the replicative lifecourse. It would therefore be helpful to determine to what degree these findings are cell type specific and which changes represent general cell senescence, especially as multiple RNAseq datasets exist on lung and skin fibroblast senescence with various mode of senescence induction. The Methods mentions 'deep senescence' but it is not clear in the main text what datasets are derived from this. Images of cells at different points along the pathway to senescence would be reassuring, and orthogonal validation of senescence markers e.g. p16, p21 by qRT-PCR is needed. It would be extremely helpful if the RNA seq and protein changes reported were cross validated by conventional techniques.

Although studies are conducted on the influence of cell density on -omics changes over a short time course of 10 days, the effect of cell cycle exit in senescence would be better accounted for by comparing with genuinely quiescent cell populations e.g. serum starvation of the hTERT cells eg Figure 3A

Time points used for comparisons are poorly labelled – are hTERT samples taken at daily intervals or at the same time points in long term culture as the RS samples? The text states that the aim is to control for the effects of long-term cell culture, but this is not apparent from later figures. Figure 1D implies that cell population at the same PDL are used for RS and hTERT groups (were cells transduced at PDL20 to allow analysis at PDL25 allowing for drug selection steps?) yet later on time points are simply given numbers e.g. 1,7 in Figure 2B and 2,4,5,6,7 in Figure 3A. It is not even stated if these are days, weeks or sample numbers.

A comparison between early PDL WT WI38 with hTERT WI38 should be conducted (the authors have the relevant data sets) to ensure that the hTERT expression simply leads to cell immortality without impacting on gene expression patterns – this is critical since telomerase has roles other than at the telomere, including in gene expression control, RNA splicing and mitochondrial metabolism. hTERT transduction is very poorly described in the methods section so that others will not be able to reproduce the experiments: "appropriate target plasmid and packaging constructs" are completely uninformative, as is 'selected. with a selection drug". Specific details are needed.

Experiments on radiation damage to induce senescence assay relatively early time points – cell cycle exit is an early event in the DNA damage response but not indicative of senescence per se. True damage-induced senescence onset should be monitored at later time points post-damage (eg from 14 days) to avoid conflating the acute DDR with senescence-specific gene expression patterns. This could account for the large discrepancy between RS and RIS datasets here.

The oxidative phosphorylation data are interesting and consistent with known changes on senescence (though the effect on ROS generation should be considered more). However, the citation in this context of papers that refer to lipid peroxidation – a state of lipid damage – rather than physiological energy generation through β oxidation of fats. Please take care not to conflate energy generating β oxidation and damaging oxidation in the citations.

The NNMT data are fascinating and really provocative – did the authors consider including bisulfite sequencing to assess DNA methylation state in conjunction with the ATAC-seq data? The finding of CTCF upregulation is also interesting in the context of the LAD/NAD data – would 3C/Hi-C give clearer data on overall chromatin configuration? (It would be intersting to see if TADs change significantly, which would be predicted from the data here).

The manuscript has not been fully proof-read which is somewhat frustrating for the reader e.g. the first few references don't even have volume or page numbers, there is incorrect use of upper/lower case in some places (e.g. amp instead of AMP) and there are various typos throughout. Methods section has gaps/omissions e.g. 4.4.3 refers to Figure ??B, 4.5.1 refers to "(vendor?)" and 4.7.2 refers to "(Cite CIS-BP)". Tenses should be consistent.

Most worryingly, very little effort has been made to ensure the Supplementary figures are informative – figure legends are lacking, labels of figure parts are missing e.g. time points/PDLs on Figure S2, Figure S3; the interpretation that EMT (or rather FMT) is a feature of sesncence depends to some extent on the data in Figure S2 but these appear to show that there is no change of the progression from early proliferation to senescence (though hard to tell as PDLs not given)? statements such as "S phase and G2M cells were isolated" – this is wholly misleading as the cells were not isolated physically according to cell cycle stage (eg FACS), but the data were processed post-hoc to determine which cell cycle stage they most closely fit with according to gene expression patterns. Figure S8 legend states cells at PDL20, figure labels cells at PDL25; time points/cell ages are missing for Figure S9- and cannot state mitochondrial functions are already simply by looking at protein levels – safer to say that changes are consistent with altered mitochondrial metabolism. Again, PDL20 is referred to yet throughout the paper, the earliest PDL analysis is at 25. Figure S10 is uninterpretable as no X axis is given, and time points/PDLs are also missing in Figure S11B. Figure S12D – what do the boxes represent? The lack of labels and suitable legend make it impossible to interpret. Why analyse PDL 46 here, as cells are not yet senescent? Is the whole of chromosome 22 shown in part E? Please provide Mb scale – and state what platform was used to view (e.g. IGV?).Figure S14 – GO terms would be a better way of showing the processes than the word clouds used. Figure S15 lacks a legend – myofibroblast markers (from lit) – I presume from literature? Which are the fibrillar collagens (the collagen number matters).

When describing what samples the data shown are compared against, the terminology "the first sample" needs more information – is this PDL25 for RS? But this 'first sample' appears later in metabolomic analysis eg Figure 3E (hence fold changes will not be comparable with proteomics).

Overall this could be a really great paper, bringing together a range of potent -omics techniques to study replicative senescence at high resolution cross time. Most of the paper is well written and provides a clear and sensible discussion of the work carried out, the data and implications of the results. It is somewhat dense (which may be inevitable with multi-omics studies) and could benefit from more accessible explanation of some of the analyses performed (eg readers will be more familiar with tSNE than UMAP – a brief explanation would help here). However, the paper is let down in parts by sloppy presentation, lack of critical experimental detail (meaning the work cannot be reproduced by others), poor design (or one hopes simply poor description) of comparators, an apparent lack of understanding of the nuances of senescent cell behaviour in culture, and by lack of statical validation of the simpler experiments (no n values noted in some cases – e.g. Figure 1D, Figure 5A, C, Figure S3 etc etc). Confirmatory experiments to cross validate top hits from the -omics studies are also required. Mining the existing proteomics data for PTMs should also be possible and may add some value, though not essential here.

*Reviewer #3 (Recommendations for the authors):*

1. There are some technical points about ATAC-seq that need to be clarified. (a) global chromatin accessibility may be higher in senescent cells. Could the authors share the bioanalyzer profiles of the ATAC-seq libraries from the different timepoints? Are they similar? (b) senescent cells typically produce far more mitochondrial reads. Could the authors confirm that sufficient depth of sequencing was achieved in their ATAC-seq runs? These data including sequencing statistics, peak coordinates, fragment distribution graphs etc. should be included in the Supplementary Section.

2. The ATAC-seq data might benefit from a RepEnrich analysis to identify peaks in repeat elements that are known to be heterochromatinized in proliferating cells and derepressed in senescence. This, in my opinion, is a better way to look at constitutive heterochromatin desilencing compared to NAD/LAD overlap and is also independent of NAD/LAD calls in a different cell line (in this case IMR-90).

3. What are the authors thoughts about the decrease in chromatin accessibility at promoters, enhancers, and gene bodies in their ATAC-seq data (Figure 5B)? Does this imply a global shut down of transcription? Is it reflected in the transcriptomic analyses?

4. While the authors perform an in-depth analyses of transcription factors most likely to drive RS, they haven't really shown the direct binding of these factors to chromatin in RS (and not hTERT, RIS or CD cells) and/or the change in their expression in senescence. Additionally, no functional experiments are performed by knocking down or overexpressing TEAD1 and investigating its effect on senescence. If possible, it would be nice to include this data (i.e., a western blot, genomic binding of TEAD1 and some functional experiments).

---

## [Author Response]

Essential revisions:1) All three reviewers have raised various concerns about robustness of analyses and statistics for the multi-omics data. They recognize that significant amounts of these analyses will need to be redone and some additional analyses were recommended. However, they all agree that these revisions are essential. Please address these concerns as much as possible.

We have preformed a statistical reanalysis of the metabolic and proteomic data sets as per the reviewers requests. In addition we have revamped the ATAC-seq analysis using limma-voom, which is compatible with quantile normalized count data to make significance calls on our ATAC-seq peak data. Lastly, we have employed higher stringency thresholds for many of our other analyses including the differential calls on our bulk RNA-seq and GSEA analysis.

2) Reviewers #2 and #3 pointed out the necessity to provide some biological validations for novel findings from these multi-omics analyses. This will require additional experiments but will greatly improve this manuscript.

We embarked on this project to create a high resolution definition of replicative senescence across multiple modalities wherein the integration of data types would foment novel hypothesis that would not have likely been made looking at any one or two data types (e.g. NNMT→metabolism →chromatin). The effort in collecting, analyzing and integrating all the modalities presented was immense, and we argue that the manuscript stands as a complete piece of work in its own right.

That being said, we wholeheartedly agree that the functional verification of the central regulatory hypothesis would clearly illustrate the utility of this data set.

To that end we have tested the central regulatory hypothesis, that the YAP1/TEAD1 transcription factor activity regulates the EMT/FMT signature in old and senescent WI-38 cells that were isolated in our experiments. Using verteporfin, a molecule that blocks YAP1/TEAD1 activity, we found that inhibition of YAP1/TEAD1 activity robustly down-regulates the published YAP targets in addition to the EMT/FMT signature in older WI-38 cells (Figure 8 and Figure 8—figure supplement 1).

In addition, we have two new analyses based on insightful reviewer questions and observations. First, reviewer number 2 pointed out that the gradual effect of increasing PDL on the transcriptome could arise from earlier onset (prior to sampling) contact inhibition due to the larger size of older cells. To test this hypothesis we scored our single cell data with significant genes that change only during replicative senescence, or only during contact inhibition. We found that only the senescence score increases with PDL suggesting that cell density is not a major confounding factor in our analysis (Figure 2—figure supplement 4).

Second, Reviewer 3 asked if the decrease in accessibility within euchromatin was indicative of a global shut down of transcription. Given that we did observe gene induction and increased accessibility in NADs, we asked if we subdivide the genome into NADs or NAD excluded regions if the chromatin state changes suggest a global shutdown. We found an increase in accessibility with replicative senescence in enhancer and alternate promoter states that reside with NADs (Figure 5—figure supplement 5) This result suggests transcription is not globally inhibited.

In all, these experiments, data, and analyses make the manuscript much stronger and much more compelling in the context of current work at the interface of wound healing, myofibroblast biology, and replicative senescence. We greatly appreciate the reviewers comments and suggestions.

3) All three reviewers have identified the lack of description for certain methods and techniques, mislabeling, typographical errors, etc. Please address all these issues.

We apologize for typos and oversights. We have fleshed out methods and fixed typos.

4) Reviewer #2 pointed out that the proteomics data should be made publicaly available at one of the major proteomics data repositories. Reviewer #1 also pointed out that in-house software codes and scripts for data analyses should also be made publicly available.

The Protemics data has been uploaded to ProteomeXchange Consortium via the PRIDE partner repository with the dataset identifier PXD02879.

Additionally all code for figures and data processing can be found here at github repository: https://github.com/dghendrickson/hayflick

Again, we thank the reviewers for their time and thoughts and appreciate their contributions to the revised manuscript.

Reviewer #1 (Recommendations for the authors):1. There are major concerns about the statistics in the analysis of the dataset.a. For the differential expression analysis (4.2.3), the authors state: "DEGs were defined as having p-values < 0.001." This is an incorrect method of filtering DEGs as it does not correct for multiple hypothesis testing. This is all the more surprising because DESeq2, used by the authors, natively provides FDR corrected p-values. The data should be re-analyzed after proper multiple hypothesis correction control (native DEseq2 FDR, Bonferroni, or other).b. Potentially related to the absence of FDR thresholding, the number of DEGs identified-"8969 genes change with increasing PDL … 11652 genes vary with increasing cell density"-seemed, to this reviewer, really high, and likely the result of improper FDR correction. It will be interesting to see how those numbers change with the multiple testing correction.

We have switched to using adjusted p-values generated by DESeq2. We have fixed this oversight in the text as well as methods. The number of significant genes is as follows: 6995 replicative senescence DEGs, 7065 radiation induced senescence DEGs, and 9958 cell density DEGs.

c. Based on the methods, the other analyses should also always incorporate a multiple testing correction, i.e. differential ATACseq peaks (4.4.3 p < 0.001, no FDR correction), metabolomics (4.5 no information on differential analysis) and proteomics (4.6.5 no information on differential analysis).

As per Reviewer 1’s request (see below), we re-analyzed the data with limma-voom and used corrected p-values for differential analysis.

For the metabolites, we have applied a linear model across time for each metabolite in both the hTERT cells and WT WI-38 cells. We now have this analysis described in the methods and include a supplementary data file with the resulting p-values and the FDR adjusted p-values [Storey and Tibshirani, 2003]. We now call out significantly changing metabolites (q-value *<* 0.05) with asterisks in relevant figures and supplementary figures.

Similarly to the metabolomics, we applied a linear model across time for each protein in both the hTERT cells and WT WI-38 cells. We have now described this analysis in the methods and include a supplementary data file with the resulting p-values and q-values. We also now mark significantly changing metabolites and proteins (adj. p value *<* 0.01) with asterisks in relevant figures and supplementary figures.

d. When carrying out differential analyses, the authors should always avoid filtering on fold change. They do this for instance in section 2.4 "we focused on compounds exhibiting PDL-dependent changes in abundance > log2 0.5.". Fold-change filtering leads to extremely poor FDR control and should not be used post hoc – the only acceptable method to integrate fold change filtering includes a new statistical framework which takes into account fold change for FDR calculations (see PMC2654802). All results should be presented with this criterion removed or re-calculated with a method that takes fold-change into account.e. Based on figure legends (Figure 1F, 3A, etc) and methods (4.2.5), FDR correction for multiple testing was also not performed for GSEA results. Please recompute everything including the correction, and only report corrected p-values.

As suggested by the reviewer, the described analysis and significance thresholds have been applied to the metabolomics and proteomics data. The p-values used in the GSEA analysis are Benjamini-Hochberg FDR adjusted pvalues. We have made this clear in text, figure legends and methods. We apologize for the confusion. We thank the reviewer for their attention to detail, and we believe the analyses are now more robust and the conclusions stronger.

f. There are several key issues with the ATAC-seq analysis (4.4.2), which will require a ground-up reanalysis of the data. Quantile normalization of count data and subsequent use of limma for differential accessibility analysis of ATAC-seq (4.4.2, end) is potential problematic. Indeed, limma is not appropriate for count data (it should only be used for continuous data such as microarray) unless the authors mean that they used limma-voom, which is a different algorithm. Please correct this in the revised analyses. In cases where differential depth cannot be corrected for appropriately, the authors should instead use downsampling of mapped reads to match the lowest depth sample. This will remove sequencing depth biases and allow proper count based methods, such as DEseq2, to perform properly.

We have reanalyzed the ATAC-seq data using limma-voom as opposed to limma. In addition, limma-voom, is compatible with quantile normalization as previously described by the voom devloper in Law et al., and others [Shi et al., 2015, Law et al., 2014]. The current analysis using the original limma significance calls on quantile normalized ATAC data have been replaced with the limma-voom significance calls on quantile normalized ATAC-seq data (manuscript Figure 6A and 6B).

2. Additional methodological information is needed for long-term reproducibility of analyses.a. For reproducibility of code and analyses, all analytical code for this study should be deposited in a repository such as github or made available as a Supplementary file.b. Please include all version numbers for all used software (e.g. R, Trimmomatic, Salmon, LISA, etc.) packages and R packages (e.g. DESeq2, fgsea, etc.) command parameters where relevant (such as for Trimmomatic, if not available elsewhere). The same should be applied to annotation databases, and if a version number doesn't exist, date of access should be provided.

All code for data processing, figure analysis and plotting, package version history and annotation database version number is now hosted in a github repository:

https://github.com/dghendrickson/hayflick with the appropriate input data files.

Command parameters and annotation database versions and software versions have also been included in methods.

c. All reagents (e.g. DMEM) should include suppliers and catalogue numbers in order to remove any ambiguity as to which product was used.

We have updated the methods section as specified and included a ”Key Reagents” table.

d. Salmon is a pseudoaligning method that uses a transcriptome reference, not a genome. The methods say that the index was derived from hg38, but how and using which information and software is not clear. In general, the authors should make sure that all key analytical steps are explicitly documented in the methods for long term reproducibility of the study.

As mentioned above this information can be located at the Git repository and methods. Specifically we have fleshed out methods sections for our pseudoalignment in manuscript.

3. It is completely unclear which data/plots were generated using the Illumina mRNA kit and which were generated using the Total RNA Kit. Please specify in text and legend.Reviewer #2 (Recommendations for the authors):The -omics aspects of the paper are generally well conducted and provide either confirmatory data on pre-existing studies of senescence or add new data (especially on the key FMT pathway and transcription factors involved). Log2FC of +/- 0.5 is somewhat lower than usual cut-offs, so justification is needed where that is selected.

As mentioned in response to reviewer 1, we have revamped the differential testing for metabolomics and proteomics data with higher stringency and FDR adjusted p-values.

No mention is made of quality control steps in RNAseq analysis, but I assume these were conducted?

We have now included a supplementary data file with QC metrics of percent reads aligning to the transcriptome and total number of reads mapped. We included a note in methods that all RNA samples had RIN scores of higher than 7 going into library preparation.

Use of in-house programs for proteomics analysis does mean that others will not be able to conduct the same analyses, though availability of datasets should allow for validation of findings on other data analysis platforms – have these been publicly deposited as links provided are only to RNAseq datasets?

Proteomics methods have been updated to cite the methods used and provide links to code for analysis. Furthermore, data has been deposited to the ProteomeXchange Consortium via the PRIDE partner repository with the dataset identifier PXD028799.

From the protocol provided, it is also not clear that the methodology for protein extraction would release chromatin-bound factors – inclusion of a potent nuclease e.g. Benzonase, would ensure that tightly bound proteins are not lost in the spin steps to remove 'cellular debris'.

We understand and agree with the reviewers point. We will take into consideration for future work.

Figure 2E: there appears to be an aberrant pattern for cells at PDL33 that is present in all parts – is 33 an outlier? What is this cluster?

We compiled the genes and ran in GoTerm finder: https://david.ncifcrf.gov/home.jsp and found these enriched terms and keywords (top 10 see below). Having not found a clear cohesive set of enrichments we think this cluster may represent noise.

Phosphoprotein, Acetylation, GO:0005515 protein binding, Nucleus, GO:0005737 cytoplasm, GO:0005829 cytosol, Isopeptide bond, GO:0005634 nucleus, hsa04071:Sphingolipid signaling pathway,Cytoplasm

The cell biology aspects are much less well described making it impossible for others to reproduce the work – in particular, cell seeding density is critical but not even mentioned for longitudinal culture, and Figure 1D images suggest significant cell crowding. Along the senescence trajectory, primary fibroblasts grown in monolayers increase in surface area (and to some extent in volume) and populations at later PDLs undergo contact inhibition at lower cell densities than those at earlier PDLs so it may be the case that some of the 'gradual' changes (especially those associated with cell cycle changes) reflect simply quiescence form contact inhibition rather than steps to senescence. Methodology to count cells and calculate PDL is needed as different labs may conduct this differently. Daily microscopy inspection of cultures to assess appropriate time points for subculturing is essential – simply choosing to split at 4 or 7 days according to PDL does not account for the morphological changes that occur across the replicative lifecourse.

We appreciate the reviewer’s attention to detail. We have added more detailed descriptions to our methods section for cell culture including the equation we used to calculate PDL. We have also added a supplementary figure with representative cell density images at time of harvest. We have also added a supplementary methods document outlining the extensive testing we carried out prior to initiation of the time course and the reasoning behind the sampling windows chosen. In our preliminary experiments testing the effect of cell density on gene expression we found that cells reach contact inhibition quicker at younger PDLs as evidenced by inhibition of cell cycle genes and up-regulation of the hypoxia response (Figure 1—figure supplement 5).

The reviewer raises an insightful point with respect to increased cell size that we had not ruled out with respect to the gradual increase in the senescent gene expression program. Although great care was taken to control cell density this is entirely plausible. Accordingly, we have added the following text, analysis and updated supplementary figure. Below is new in manuscript:

“It has previously been reported and observed here (Figure 2—figure supplement 5) that cell size increases with PDL in primary cell lines [Ogrodnik et al., 2019]. We hypothesized that perhaps the higher PDL cells by nature of their larger size, might reach contact inhibition faster than early PDL cells. In this scenario, given the high correlation between the gene expression signatures of senescence and cell density, the increase in senescence signature could be an artifact of cell culture arising from changing morphology. To test this possibility, we scored our single cells again with modified signatures for senescence and cell density composed only of genes uniquely induced in either perturbation (Figure 2—figure supplement 4). We found that the specific senescent signature increases with PDL in WT WI-38 cells in contrast with the specific cell density signature which stays relatively flat providing additional evidence for a gradual increase in the senescent gene expression program that is independent from changes in cell density.”

The following text is the additional supplementary methods.

“0.1 Sampling proliferating WI-38 cells at different cell densities confounds assessment of cell cycle state and senescence

A limited pilot (performed prior to RS time course) experiment was conducted to determine optimal sampling conditions. Using gene expression as a readout for cellular state, we performed RNA-seq on PDL 20, PDL 29, and PDL 39 WI-38 cells. We plated an equal number of cells in triplicate for the three PDLs and cells were collected for RNA extraction after 4 days of growth. We looked for any significant changes in transcript levels between the three samples.”

We observed two notable trends. First, we found stronger expression of genes associated with cell cycle and mitosis in the older PDL 39 cells compared to PDL 20 and PDL 29 cells. This was unexpected based on previous observations of slower growth and less active mitosis in cells closer to senescence.(Hayflick 1965). Second, the cells at PDL 20 and PDL 29 expressed genes associated with hypoxic stress (Figure 1—figure supplement 5A).

These data, in tandem with the observation that older cells grow at a slower rate, suggest that the highly proliferative younger cells (PDL 20 and 29) grow to higher cell density faster and therefore encounter earlier in real time, the various normal mitosis-inhibiting stresses, e.g. hypoxia, contact inhibition, and starvation that trigger cell cycle exit (Dayan et al., 2009).

To understand the effects of growth rates and cell density on gene expression, we performed a cell density study (CD) wherein young PDL cells were sampled at increasing levels of cell density to measure the effects of cell density and proliferation on gene expression (Figure 1A). We observed that based on the seeding density we employed (to minimize the stress of sparse plating) the sampling window we had originally targeted (4 days after seeding) was located in a volatile region with respect to proliferation rate and hypoxic stress using Cyclin B1 (CCNB1) and VegfA (VEGFA) expression as proxies for each respectively (Figure 1—figure supplement 5B).

Accordingly, we modified the sampling window by choosing a time interval (2.5-3 days) prior to the onset of VEGF-A induction and CCNB1 repression (Figure 1—figure supplement 5B). For older, slower growing cells in the plateauing region of the replicative senescence process (PDL *>* 40) we shifted the sampling window to 3.5-4 days based on visual inspection and cell counts. Ultimately in the actual extended time course, we tracked cell density with a phase contrast EVOS microscope to ensure cellular confluence was similar across all time points during sampling (Figure 2—figure supplement 5).

It would therefore be helpful to determine to what degree these findings are cell type specific and which changes represent general cell senescence, especially as multiple RNAseq datasets exist on lung and skin fibroblast senescence with various mode of senescence induction.

As mentioned, we do compare our data set to a model compiled by Hernandez-Segura 2017 over multiple skin and lung fibroblast cell lines approaching replicative senescence and find high concordance (manuscript Figure 1D). Although we agree with the reviewer that it would be informative to expand this to a comprehensive analysis of all types of senescence induction, it is outside the scope of the manuscript.

The Methods mentions 'deep senescence' but it is not clear in the main text what datasets are derived from this. Images of cells at different points along the pathway to senescence would be reassuring, and orthogonal validation of senescence markers e.g. p16, p21 by qRT-PCR is needed. It would be extremely helpful if the RNA seq and protein changes reported were cross validated by conventional techniques.

We thank the reviewer for these suggestions. The ’deep senescence’ time course consists of a second time course and three samples at 45, 52, 53 PDLs that were used for bulk RNA-seq. We have added a sample manifest that clearly shows what assays were performed on which samples and time courses (Figure 1—figure supplement 1). We added the second deep senescence time course after De Cecco et al., reported transcriptomic differences between early and deep senescence to ensure we were not missing a crucial aspect of replicative senescence [De Cecco et al., 2019]. The PDL 45 cells served as a bridge sample for batch correction so that we could accurately compare the deep senescence time points to first time course. We have added clarifying language and updated manuscript Figure 1B.

Additionally, as mentioned we now have images of cells at harvest for original time course (Figure 2—figure supplement 5).

For orthogonal validation, we would point out that each different modality of ”omics” experiments validates the other orthogonally. For example, many of the genes that we highlight as induced at the transcriptomic level (e.g. NNMT or TGFB2) are also induced in our proteomics data, our single cell RNA-seq data, and have significant change in their promoter regions in our ATAC-seq data. The strength of this work in part rests with this intrinsic cross-validation across modalities. In addition, all of our experiments were performed with replicates and at adequate depth for NGS experiments.

In addition, a plethora of previous work has demonstrated that qPCR and RNA-seq data are highly correlated [Wu et al., 2014, Shi and He, 2014, Asmann et al., 2009, Griffith et al., 2010]. We argue that the orthogonal validation across multiple modalities is substantial validation and sufficient for the conclusions drawn. In addition we argue superior to qPCR in terms of satisfying orthogonal validation given that RNA-seq and qPCR are both based on reverse transcription and cDNA amplification.

We have collated the RNA-seq and proteomics data for p21, p16, and β-galactodsidase along side NNMT and TGFB2 to demonstrate the overlap between technologies (Figure 1—figure supplement 2). For p16 and p21 we have also included Western blots to confirm as the induction in the proteomics data exhibits high variance.

Although studies are conducted on the influence of cell density on -omics changes over a short time course of 10 days, the effect of cell cycle exit in senescence would be better accounted for by comparing with genuinely quiescent cell populations e.g. serum starvation of the hTERT cells eg Figure 3A

Similar to comparing replicative senescence to other senescence inductions, we agree with the reviewer that a full workup of comparing senescence to many types of cell cycle inhibition would further our understanding about the regulatory logic of cell cycle exit and what may or may not be specific to replicative senescence. However, our aim in using the confluence condition was to account for cell cycle inhibition driven by cell density as we reasoned it was the most likely source of variability in our experimental set up. A comprehensive comparison to multiple types of cell cycle exit, although an important data set, is outside the scope of this work.

Time points used for comparisons are poorly labelled – are hTERT samples taken at daily intervals or at the same time points in long term culture as the RS samples? The text states that the aim is to control for the effects of long-term cell culture, but this is not apparent from later figures. Figure 1D implies that cell population at the same PDL are used for RS and hTERT groups (were cells transduced at PDL20 to allow analysis at PDL25 allowing for drug selection steps?) yet later on time points are simply given numbers e.g. 1,7 in Figure 2B and 2,4,5,6,7 in Figure 3A. It is not even stated if these are days, weeks or sample numbers.

We apologize for the confusing description. We have added clarifying text to emphasize that the hTERT timepoints are matched to WT PDL samples. We have also added a visual dotted line connection in manuscript Figure 1B to highlight that the samples are paired. The transduction to generate the WI-38 hTERT cells happened far in advance of the time course. We have updated the methods to reflect this.

A comparison between early PDL WT WI38 with hTERT WI38 should be conducted (the authors have the relevant data sets) to ensure that the hTERT expression simply leads to cell immortality without impacting on gene expression patterns – this is critical since telomerase has roles other than at the telomere, including in gene expression control, RNA splicing and mitochondrial metabolism. hTERT transduction is very poorly described in the methods section so that others will not be able to reproduce the experiments: "appropriate target plasmid and packaging constructs" are completely uninformative, as is 'selected. with a selection drug". Specific details are needed.

The selection process in generating the hTERT line, has made meaningful comparison to address the specific effect of hTERT over-expression impossible. To test that, we would need an inducible hTERT system to follow cells after hTERT induction. In this study the hTERT cells were only a control for the long term culture effects and not to test the effects of hTERT expression.

We agree with the reviewer that this would be a worthwhile comparison if we had the appropriate experimental set up. We have clarified our methods concerning the generation of the WI-38 hTERT cells to make this clear. The transduction to make the hTERTs occurred far in advance of the time course.

Experiments on radiation damage to induce senescence assay relatively early time points – cell cycle exit is an early event in the DNA damage response but not indicative of senescence per se. True damage-induced senescence onset should be monitored at later time points post-damage (eg from 14 days) to avoid conflating the acute DDR with senescence-specific gene expression patterns. This could account for the large discrepancy between RS and RIS datasets here.

We agree with the reviewer. We have changed the language surrounding comparisons to the radiation induced senescence condition and have added clarifying context where relevant and in the discussion.

The oxidative phosphorylation data are interesting and consistent with known changes on senescence (though the effect on ROS generation should be considered more). However, the citation in this context of papers that refer to lipid peroxidation – a state of lipid damage – rather than physiological energy generation through β oxidation of fats. Please take care not to conflate energy generating β oxidation and damaging oxidation in the citations.

We apologize we are not sure which reference the reviewer is referring to. The only paper with ”lipid peroxidation” in the title we cited was Flor et al., 2017, A signature of enhanced lipid metabolism, lipid peroxidation and aldehyde stress in therapy-induced senescence [Flor et al., 2017]. We cited this paper in the following section summing up our metabolic observations.

Indeed, metabolomic data collected from various types of senescence models has shown increased lipid oxidation, lipid accumulation, TCA up-regulation, and glycolytic alterations [Flor et al., 2017].

We cited Flor et al., in reference to increased lipid oxidation as pertaining to fatty acid β-oxidation as fatty acids are common components of complex lipids. Flor et al., observed increased β-oxidation in Figures 1A,B, C and Figure 4 with therapy induced senescence. To remove any confusion we have changed the text to read ”shown increased fatty acid oxidation, lipid accumulation, TCA up-regulation, and glycolytic alterations”

The NNMT data are fascinating and really provocative – did the authors consider including bisulfite sequencing to assess DNA methylation state in conjunction with the ATAC-seq data? The finding of CTCF upregulation is also interesting in the context of the LAD/NAD data – would 3C/Hi-C give clearer data on overall chromatin configuration? (It would be intersting to see if TADs change significantly, which would be predicted from the data here).

We agree with the reviewer and find this result very exciting. We are in fact working on directly measuring DNA methylation chromatin conformation during replicative senescence but as part of a future publication related to the findings here.

The manuscript has not been fully proof-read which is somewhat frustrating for the reader e.g. the first few references don't even have volume or page numbers, there is incorrect use of upper/lower case in some places (e.g. amp instead of AMP) and there are various typos throughout. Methods section has gaps/omissions e.g. 4.4.3 refers to Figure ??B, 4.5.1 refers to "(vendor?)" and 4.7.2 refers to "(Cite CIS-BP)". Tenses should be consistent.

We apologize and have gone through the manuscript to fix typos.

Most worryingly, very little effort has been made to ensure the Supplementary figures are informative – figure legends are lacking, labels of figure parts are missing e.g. time points/PDLs on Figure S2, Figure S3; the interpretation that EMT (or rather FMT) is a feature of sesncence depends to some extent on the data in Figure S2 but these appear to show that there is no change of the progression from early proliferation to senescence (though hard to tell as PDLs not given)?

We have remade many of the supplementary figures and main figures with more labels. The GSEA enrichment can be hard to interpret with respect to magnitude of change, i.e. even in the face of monotonic increase, the p-value of GSEA enrichment would not necessarily go up as the analysis is based on rank rather than magnitude. We thank the reviewer for calling this out as it is central to the EMT/FMT conclusions drawn. We have updated the figure to include the actual log2 fold changes of the genes driving the enrichment at each PDL to emphasize the progression from early proliferation to senescence (Figure 1—figure supplement 4).

statements such as "S phase and G2M cells were isolated" – this is wholly misleading as the cells were not isolated physically according to cell cycle stage (eg FACS), but the data were processed post-hoc to determine which cell cycle stage they most closely fit with according to gene expression patterns.

We agree with the reviewer and have fixed the language.

Figure S8 legend states cells at PDL20, figure labels cells at PDL25; time points/cell ages are missing for Figure S9- and cannot state mitochondrial functions are already simply by looking at protein levels – safer to say that changes are consistent with altered mitochondrial metabolism. Again, PDL20 is referred to yet throughout the paper, the earliest PDL analysis is at 25.

We have added a label key for the supplementary figure. Reference time points for log2 fold change heatmaps are not shown as values are at or near (when mean subtracted) zero. We have emphasized what is being compared (e.g. vs. PDL20 rather than first timepoint) and what is not being shown in figure legends.

Figure S10 is uninterpretable as no X axis is given, and time points/PDLs are also missing in Figure S11B. Figure S12D – what do the boxes represent? The lack of labels and suitable legend make it impossible to interpret. Why analyse PDL 46 here, as cells are not yet senescent? Is the whole of chromosome 22 shown in part E? Please provide Mb scale – and state what platform was used to view (e.g. IGV?).Figure S14 – GO terms would be a better way of showing the processes than the word clouds used. Figure S15 lacks a legend – myofibroblast markers (from lit) – I presume from literature? Which are the fibrillar collagens (the collagen number matters).

We have added PDL and timepoint labels to all figures. The long arm of Chr22 is shown and that is reflected in figure now.

PDL 46 was used as the changes in epigenome we report begin prior to PDL 46. As the observed heterochromatic opening begins prior to senescence, there was not a requirement to use senescent cells. If these changes were driven by dead cells, then treatment of cells with PMA prior to ATAC-seq would have altered the results in PDL 46.

The ATAC-seq supplementary figure was rendered in R. We have the relevant packages listed and code is now accessible. The heatmap from the pseudotime clustering was made using 800 GO terms that cannot fit into figure. We have now included the data underlying the heatmap for readers who would like to see the GO terms that compose the word clouds.

We have removed Figure S15 in the revised manuscript- However Col5A2 was labeled in the figure.

When describing what samples the data shown are compared against, the terminology "the first sample" needs more information – is this PDL25 for RS? But this 'first sample' appears later in metabolomic analysis eg Figure 3E (hence fold changes will not be comparable with proteomics).

We have added more explicit labels and included a sample manifest (Figure 1—figure supplement 1) to make it easier for readers to see what the time points are available for each modality. It is true that we do not have the exact same time course samples for all modalities e.g. metabolomics and proteomics.

However, we did not, and do not want readers to directly compare fold changes. Even if both metabolomics and proteomics had the exact same reference point, there is not an expectation that any of the modalities should scale similarly in magnitude or time. The aim of the figures is to show change over time from proliferating state to replicative senescence. It was extraordinarily hard to seed and sample the cells required for these experiments. When we lost the PDL20 sample for metabolomics and single cell RNA-seq, we opted not to throw out PDL20 for all other modalities as the change over time with respect to proliferation state was the key parameter and PDL25 and PDL20 are both very much in the proliferative state (manuscript Figure 1B).

Overall this could be a really great paper, bringing together a range of potent -omics techniques to study replicative senescence at high resolution cross time. Most of the paper is well written and provides a clear and sensible discussion of the work carried out, the data and implications of the results. It is somewhat dense (which may be inevitable with multi-omics studies) and could benefit from more accessible explanation of some of the analyses performed (eg readers will be more familiar with tSNE than UMAP – a brief explanation would help here).

We thank the reviewer for praise and appreciate the criticisms which helped improve the manuscript overall. We have added a quick UMAP explanation in the manuscript.

However, the paper is let down in parts by sloppy presentation, lack of critical experimental detail (meaning the work cannot be reproduced by others), poor design (or one hopes simply poor description) of comparators, an apparent lack of understanding of the nuances of senescent cell behaviour in culture, and by lack of statical validation of the simpler experiments (no n values noted in some cases – e.g. Figure 1D, Figure 5A, C, Figure S3 etc etc).

Hopefully, the changes we have made will strengthen the paper. Specifically, we hope that the additional methods and supplementary text concerning cell culture and hTERT cell line generation convince the reviewer that we took great care in our design and cell culture. As mentioned, we specifically chose cell density as a control as cell density is most likely to be altered between sampling periods especially in cells changing in size and growth rate. For example, the cell density experiment combined with the single cell data bolsters our conclusion of ’gradual’ onset senescence and provides evidence beyond that our careful work with respect to sampling density was successful (Figure 2—figure supplement 4).

We have added statistical validation of simpler experiments in manuscript Figure 1. manuscript Figure 5A was an exploratory analysis that was simply a starting point for manuscript Figure 5B, i.e. we would have proceeded with higher resolution analysis even if global percentage changes were not significant. The significance testing for the analysis in manuscript Figure 5C, originally just in text, has been added to the figure.

Confirmatory experiments to cross validate top hits from the -omics studies are also required. Mining the existing proteomics data for PTMs should also be possible and may add some value, though not essential here.

As mentioned, we argue that the different modalities serve as sufficient cross validation and that experiments to confirm changes that are robust across multiple disparate data types would not add to conclusions or invalidate them. For example, an inconsistency from a western blot of NNMT could signify that the purchased antibody has poor binding or specificity since we have available data to validate from the gene level (RNA-seq), protein level (proteomics), and readout of cellular metabolic state (metabolomics). A simple western blot or qPCR would not strengthen the conclusions further. However as mentioned above, we have included p21 and p16 due to variance in proteomics data.

Reviewer #3 (Recommendations for the authors):1. There are some technical points about ATAC-seq that need to be clarified. (a) global chromatin accessibility may be higher in senescent cells. Could the authors share the bioanalyzer profiles of the ATAC-seq libraries from the different timepoints? Are they similar? (b) senescent cells typically produce far more mitochondrial reads. Could the authors confirm that sufficient depth of sequencing was achieved in their ATAC-seq runs? These data including sequencing statistics, peak coordinates, fragment distribution graphs etc. should be included in the Supplementary Section.

A) We have added a supplementary figure with representative bionalyzer traces (Figure 5—figure supplement 1 and 2). ATACseq libraries from all PDLs sampled share the same fragment length distribution before and after sequencing. We employed a size selection cut-off to enrich for nucleosome free regions.

B) We have added a supplementary figure with mitochondrial percentages and transcription start site enrichment (Figure 9). We do not observe an increase in mitochondrial percentage (less than 2% across all samples) with increasing PDL. We attribute the low mitochondrial percentages to the use of the Corces et al., ”Omni-ATAC” protocol, which claims a 13-fold decrease in reads that map to mtDNA compared to the standard ATAC-seq protocol [Corces et al., 2017].

In addition, we have included a supplementary file with read depth, duplicate read percentages, mitochondrial read percentage, alignment statistics, and TSS score for each sample–all of which pass the ENCODE pipeline standards: https://www.encodeproject.org/atac-seq/.

Significant peak coordinates were included in original manuscript and in revised manuscript. They can also be found in GEO repository along with bedgraphs of alignment and coverage. We included representative fragment length distribution plots which mirror the size selected ATAC-libraries sequenced (Figure 5—figure supplement 1, final pool bioA trace).

2. The ATAC-seq data might benefit from a RepEnrich analysis to identify peaks in repeat elements that are known to be heterochromatinized in proliferating cells and derepressed in senescence. This, in my opinion, is a better way to look at constitutive heterochromatin desilencing compared to NAD/LAD overlap and is also independent of NAD/LAD calls in a different cell line (in this case IMR-90).

We appreciate the reviewer’s suggestion. However, for our analysis we specifically were interested in experimentally validated NAD and LAD domains which contain both gene bodies and heterochromatin as opposed to solely constitutive heterochromatin (see point 3 below). We argue that use of the IMR-90 labels is justified based on the observed decrease in transcription start site coverage in Figure 5—figure supplement 3. These analyses was carried out solely based on genomic gene annotation independent of cell type. That we observe the similar results when using chromatin state labels derived from IMR-90 cells (also fetal lung fibroblasts) independent of gene annotation (manuscript Figure 5A and 5B) suggests that the IMR90 derived labels are accurately reflecting WI-38 chromatin state.

We think that digging deeper into constitutive heterochromatin especially in light of our NNMT findings is an exciting avenue for further research and is part of ongoing follow up work, and we will look into RepEnrich analysis.

3. What are the authors thoughts about the decrease in chromatin accessibility at promoters, enhancers, and gene bodies in their ATAC-seq data (Figure 5B)? Does this imply a global shut down of transcription? Is it reflected in the transcriptomic analyses?

This is an important question. We interpret the results as a winnowing down or specialization of function in senescent cells (e.g. permanent cell cycle exit, etc. ). Although the aggregate pattern is down, we do observe promoters that exhibit increased accessibility resulting in higher transcription of neighboring genes and increased protein levels e.g. NNMT and TGFB2. In fact, in response to this question, we carried out the same analysis used in the manuscript Figure 5B (changing peak accessibility by chromatin state label) and instead divided into peaks overlapping or excluded from NADs (Figure 5—figure supplement 5). We found that ATAC-seq peaks within NADs exhibit an altered chromatin state profile. For NAD exluded peaks, the results largely mirror the results for all peaks (increased accessibility in closed regions and vice versa). Conversely, multiple chromatin states in the enhancer and promoter categories, which decrease in accessibility with PDL in NADs excluded peaks, display increased accessibility with increasing PDL only in NAD overlapping peaks.

Given that we also observe overlap with replicative senescent gene expression from bulk RNA-seq data and NADs (manuscript Figure 5C), we interpret this result as consistent with the hypothesis that transcription is not entirely shutting down rather so much as changing where it is happening. Basically, although widespread, the reduced accessibility in euchromatin observed in aggregate, is not reflective of a pervasive shutdown of transcription across the entire genome.

We appreciate the reviewer’s question. We have added this result as a supplement to the main text.

4. While the authors perform an in-depth analyses of transcription factors most likely to drive RS, they haven't really shown the direct binding of these factors to chromatin in RS (and not hTERT, RIS or CD cells) and/or the change in their expression in senescence. Additionally, no functional experiments are performed by knocking down or overexpressing TEAD1 and investigating its effect on senescence. If possible, it would be nice to include this data (i.e., a western blot, genomic binding of TEAD1 and some functional experiments).

We embarked on this project to create a high resolution definition of replicative senescence across multiple modalities wherein the integration of data types would incite novel hypothesis that would not have likely been made looking at any one or two data types (e.g. NNMT →metabolism →chromatin). The effort in collecting, analyzing and integrating all the modalities presented was extensive, and we argue that the manuscript stands as a complete piece of work in its own right. However, we do agree that the functional verification of the central regulatory hypothesis would clearly illustrate the utility of this data set.

To that end, we have taken late passage PDL 40 WI-38 cells wherein increase in expression of the EMT signature was observed, (Figure 5B) and treated with verteporfin, a molecule that blocks YAP1/TEAD1 interaction as demonstrated by [Mascharak et al., 2021]. We preformed RNA-seq on these treated cells and found robust inhibition of known YAP targets, TGFB2, and the EMT signature that started this line of inquiry (Figure 8—figure supplement 1). These data are consistent with a regulatory role for YAP1/TEAD1 in establishing the EMT/FMT gene expression program in WI-38 cells approaching replicative senescence.

These data make the paper stronger and highlight the utility and relevance of the generated data sets. We thank the reviewer for this suggestion.

References

Asmann et al., 2009 Asmann, Y. W., Klee, E. W., Thompson, E. A., Perez, E. A., Middha, S., Oberg, A. L., Therneau, T. M., Smith, D. I., Poland, G. A., Wieben, E. D., et al. (2009). 3’tag digital gene expression profiling of human brain and universal reference rna using illumina genome analyzer. *BMC genomics*, 10(1):1–11.

Corces et al., 2017 Corces, M. R., Trevino, A. E., Hamilton, E. G., Greenside, P. G., Sinnott-Armstrong, N. A., Vesuna, S., Satpathy, A. T., Rubin, A. J., Montine, K. S., Wu, B., et al. (2017). An improved atac-seq protocol reduces background and enables interrogation of frozen tissues. *Nature methods*, 14(10):959–962.

De Cecco et al., 2019 De Cecco, M., Ito, T., Petrashen, A. P., Elias, A. E., Skvir, N. J., Criscione, S. W., Caligiana, A., Brocculi, G., Adney, E. M., Boeke, J. D., et al. (2019). L1 drives ifn in senescent cells and promotes age-associated inflammation. *Nature*, 566(7742):73–78.

Flor et al., 2017 Flor, A. C., Wolfgeher, D., Wu, D., and Kron, S. J. (2017). A signature of enhanced lipid metabolism, lipid peroxidation and aldehyde stress in therapy-induced senescence. *Cell death discovery*, 3(1):1–12.

Griffith et al., 2010 Griffith, M., Griffith, O. L., Mwenifumbo, J., Goya, R., Morrissy, A. S., Morin, R. D., Corbett, R., Tang, M. J., Hou, Y.-C., Pugh, T. J., et al. (2010). Alternative expression analysis by rna sequencing. *Nature methods*, 7(10):843–847.

Hernandez-Segura et al., 2017 Hernandez-Segura, A., de Jong, T. V., Melov, S., Guryev, V., Campisi, J., and Demaria, M. (2017). Unmasking transcriptional heterogeneity in senescent cells. *Curr. Biol.*, 27(17):2652–2660.e4.

Law et al., 2014 Law, C. W., Chen, Y., Shi, W., and Smyth, G. K. (2014). voom: Precision weights unlock linear model analysis tools for rna-seq read counts. *Genome biology*, 15(2):1–17.

Mascharak et al., 2021 Mascharak, S., Davitt, M. F., Griffin, M., Borrelli, M. R., Moore, A. L., Chen, K., Duoto, B., Chinta, M., Foster, D. S., Shen, A. H., et al. (2021). Preventing engrailed-1 activation in fibroblasts yields wound regeneration without scarring. *Science*, 372(6540).

Ogrodnik et al., 2019 Ogrodnik, M., Salmonowicz, H., Jurk, D., and Passos, J. F. (2019). Expansion and cell-cycle arrest: common denominators of cellular senescence. *Trends in biochemical sciences*, 44(12):996–1008.

Shi et al., 2015 Shi, W., Liao, Y., Willis, S. N., Taubenheim, N., Inouye, M., Tarlinton, D. M., Smyth, G. K., Hodgkin, P. D., Nutt, S. L., and Corcoran, L. M. (2015). Transcriptional profiling of mouse b cell terminal differentiation defines a signature for antibody-secreting plasma cells. *Nature immunology*, 16(6):663–673.

Shi and He, 2014 Shi, Y. and He, M. (2014). Differential gene expression identified by rna-seq and qpcr in two sizes of pearl oyster (pinctada fucata). *Gene*, 538(2):313–322.

Storey and Tibshirani, 2003 Storey, J. D. and Tibshirani, R. (2003). Statistical significance for genomewide studies. *Proceedings of the National Academy of Sciences*, 100(16):9440–9445.

Wu et al., 2014 Wu, A. R., Neff, N. F., Kalisky, T., Dalerba, P., Treutlein, B., Rothenberg, M. E., Mburu, F. M., Mantalas, G. L., Sim, S., Clarke, M. F., et al. (2014). Quantitative assessment of single-cell rna-sequencing methods. *Nature methods*, 11(1):41–46.